# Synthesis and Anti-Hepatocarcinoma Effect of Amino Acid Derivatives of Pyxinol and Ocotillol

**DOI:** 10.3390/molecules26040780

**Published:** 2021-02-03

**Authors:** Ying Zhang, Hui Yu, Shuzheng Fu, Luying Tan, Junli Liu, Baisong Zhou, Le Li, Yunhe Liu, Caixia Wang, Pingya Li, Jinping Liu

**Affiliations:** 1School of Pharmaceutical Sciences, Jilin University, Fujin Road 1266, Changchun 130021, China; zhangying17@mails.jlu.edu.cn (Y.Z.); yuhui19@mails.jlu.edu.cn (H.Y.); fusz19@mails.jlu.edu.cn (S.F.); tanly20@mails.jlu.edu.cn (L.T.); junli18@mails.jlu.edu.cn (J.L.); bszhou19@mails.jlu.edu.cn (B.Z.); lile19@mails.jlu.edu.cn (L.L.); yunhe20@mails.jlu.edu.cn (Y.L.); cxwang20@mails.jlu.edu.cn (C.W.); lipy@jlu.edu.cn (P.L.); 2The First Hospital of Jilin University, Changchun 130021, China; 3Research Center of Natural Drug, Jilin University, Changchun 130021, China

**Keywords:** pyxinol, ocotillol, amino acid derivatives, anti-hepatocarcinoma, metabolomics

## Abstract

Aiming at seeking an effective anti-hepatocarcinoma drug with low toxicity, a total of 24 amino acid derivatives (20 new along with 4 known derivatives) of two active ocotillol-type sapogenins (pyxinol and ocotillol) were synthesized. Both in vitro and in vivo anti-hepatocarcinoma effects of derivatives were evaluated. At first, the HepG2 human cancer cell was employed to evaluate the anti-cancer activity. Most of the derivatives showed obvious enhanced activity compared with pyxinol or ocotillol. Among them, compound **2e** displayed the most excellent activity with an IC_50_ value of 11.26 ± 0.43 µM. Next, H22 hepatoma-bearing mice were used to further evaluate the anti-liver cancer activity of compound **2e**. It was revealed that the growth of H22 transplanted tumor was significantly inhibited when treated with compound **2e** or compound **2e** combined with cyclophosphamide (CTX) (*p* < 0.05, *p* < 0.01), and the inhibition rates of tumor growth were 35.32% and 55.30%, respectively. More importantly, compound **2e** caused limited damage to liver and kidney in contrast with CTX causing significant toxicity. Finally, the latent mechanism of compound **2e** was explored by serum and liver metabolomics based on ultra-performance liquid chromatography quadrupole time-of-flight mass spectrometry (UPLC-QTOF-MS) technology. A total of 21 potential metabolites involved in 8 pathways were identified. These results suggest that compound **2e** is a promising agent for anti-hepato-carcinoma, and that it also could be used in combination with CTX to increase efficiency and to reduce toxicity.

## 1. Introduction

Liver cancer, the fourth major lethal malignancy worldwide, is generally divided into HCC (hepatocellular carcinoma), ICC (intrahepatic cholangiocarcinoma) and MHC (mixed hepato- cellular and cholangiocellular carcinoma) based on the pathological classification [1]. HCC, with 5-year overall survival, is only 10%, yet accounts for nearly 75% to 85% of liver cancer cases [2,3]. Despite the enormous efforts such as surgery, chemotherapy, radiation therapy, immunotherapy, and monoclonal antibody therapy to combat liver cancer, the disease burden imposed by it continues to increase [4]. Numerous scientific researchers devoted themselves to the modification or transformation of lead compounds with the purpose of obtaining derivative products with good properties [5,6,7,8]. Natural products and their structurally modified derivatives have been widely used as anti-tumor drugs due to the low toxicity and the ability of reversing multidrug resistance [9,10,11,12].

Ocotillol-type saponins or sapogenins, with a tetrahydrofuran ring in side chain, are the natural triterpenoids [13,14]. Pyxinol and ocotillol are two important sapogenins [11]. Pyxinol (CAS number: 25330-18-1) was isolated from some natural plants such as *Pyxine endochrysina* N_YL_, *Notholaena rigidain*, *Salvia barrelieri* and *Betula humilis*, respectively [15,16,17,18]. Interestingly, pyxinol is also the active metabolite of protopanaxadiol (the main aglycon of ginsenosides) in human liver [19]. It was reported that pyxinol exhibited multiple pharmacological activities such as anti-cancer (including lung carcinoma, cervical carcinoma and colonic adenocarcinoma) [20,21,22], anti-multidrug resistance [11] and anti-inflammatory [23], etc. Ocotillol (CAS number: 5986-39-0)—firstly discovered from *Panax pseudoginseng* Subsp. *Himalaicus* collected in Eastern Himalaya [24]—was also the main metabolite of ginsenosides in American ginseng [25]. Ocotillol had the effects of anti-ovarian cancer, anti-liver cancer [26,27,28,29], anti-inflammatory [30], anti-bacterial [31], etc. Our team have successfully prepared pyxinol and ocotillol with high yields by treating various ginsenosides [32]. However, in the continuing study, we found that pyxinol and ocotillol both had the unsatisfactory properties that limited their application and development. Pyxinol, with the low oral bioavailability and short *t*_1/2_ in rats [33], showed weaker activities (such as anti-heart failure, anti-inflammatory and anti-bacterial) than its derivatives in vitro and in vivo [23,31,34]. Ocotillol, with low water solubility [35] and high plasma protein binding rates [36], had the low systemic exposure, poor absorption into blood and long *t_1/2_* in rats [37]. Therefore, it is necessary to modify pyxinol and ocotillol into derivatives with better properties and better pharmacological activity. The introduction of amino acid groups in the structure could improve the solubility, enhance active transport, promote intestinal absorption, reduce side effects and improve bioavailability of natural products [38]. From all the above facts, we anticipated that amino acid conjugated ocotillol-type sapogenin might result in new leads possessing good pharmacological activities.

According to the reports, the properties or the activities of lead compounds could be improved by amino acid groups [39,40,41,42,43,44,45], and the research about the triterpenoids conjugated with amino acids has made good progress. Pyxinol or 24S-pyxinol conjugated with amino acids showed better antibacterial or better anti-inflammatory activities [46]. The amino acid derivatives of 25-OCH3/OH-Protopanaxadiol showed higher anti-tumor activity [47]. In addition, it was reported that the anti-liver tumor activity could be significantly improved by conjugating sapogenin AD-1 with amino acids [48]. Amino acids are also closely related to the reduction in the incidence of HCC in patients with liver cirrhosis [49,50]. Moreover, as an important auxiliary treatment method for the improvement of hepatopathy, appropriate supplementation of amino acids have attracted more and more attention [51].

The present study designed and synthesized a series of amino acid derivatives of pyxinol and ocotillol. HepG2 cell line was then employed to evaluate the anti-cancer activity of all the derivatives. H22 hepatoma-bearing mice were furtherly used to evaluate the activity of potential anti-tumor agent. The ultra-performance liquid chromatography quadrupole time-of-flight mass spectrometry (UPLC-QTOF-MS) based metabolomics approach, a burgeoning omics technique by measuring the dynamic multiparametric metabolic response of the living system, was finally applied to explore the latent mechanism of the active derivative.

## 2. Results and Discussion

### 2.1. Design

As for the structure of ocotillol-type sapogenins, 3-OH is relatively far from other hydrogen donor or acceptor atoms and dominates an important place in the metabolic pathway. According to the previous literature, 3-OH was suitable for modification to improve the activity without affecting other potential binding sites [52,53]. Many efforts and progress had been made in modifying 3-OH of ocotillol-type sapogenins. The substitution at C-3 hydroxy such as the oxidation products [54], aromatic carboxylate derivatives [55], alkylation products [56], nitrate derivatives [20], amino acid derivatives [28], bromoalkanoates derivatives [20], oxime derivatives [53] and N-substituted amide derivatives [57] could all significantly enhance the biological activity of ocotillol-type sapogenins. Therefore, 3-OH was selected for further modification [19] in our study.

Amino acid imbalance is related to the liver diseases. For examples: the level of alanine (Ala) was found to be significantly reduced in cirrhotic rats, [58]; the levels of leucine (Leu) and valine (Val) were also clarified to decrease in patients with liver cirrhosis [50,59]; glycine (Gly) could alleviate liver injury in rats through its antioxidant properties [60]; methionine (Met) plays a key role in many cellular functions, deficiency of Met in rats could induce liver injury [60]; proline (Pro) could scavenge free radicals and reduce tissue damage in patients with liver cirrhosis [61]. What is more, the amino acids that make up natural proteins are all L-type, which is more prone to proteolysis. So, in the present study, the typical aliphatic amino acids (Ala, Leu, Val, Gly and Met) and heterocyclic amino acid (Pro) were chosen to modify pyxinol and ocotillol (Figure 1).

### 2.2. Chemistry

A total of 24 amino acid derivatives of pyxinol and ocotillol were synthesized as shown in Figure 2. Compounds **1a**–**1f** and **3a**–**3f** were separately synthesized by pyxinol and ocotillol in the presence of EDC {1-ethyl-3-(3-dimethylaminopropyl) carbodiimide} and DMAP (4-dimethylamino- pyridine) mediated esterifications of N-Boc protected amino acids in dry CH_2_Cl_2_ or THF (tetra- hydrofuran) at room temperature. The tert-butyl carbonate groups of them were removed in a mixed solvent of dry CH_2_Cl_2_ and TFA (trifluoroacetic acid) at room temperature to obtain compounds **2a**–**2f** and **4a**–**4f**. Silica gel column chromatography was used for purification of all the derivatives from the reaction mixture, and the structures of all products were confirmed by ^1^H-NMR, ^13^C-NMR and HR-MS. Taking compound **2e** as an example, its molecular formula was established as C_35_H_59_NO_5_ by HR-ESI-MS. Compared with pyxinol (C_30_H_52_O_4_, the lead compound), compound **2e** has an increased part (C_5_H_7_NO, equivalent to the deshydroxyproline). The ^1^H-NMR (CDCl_3_, 600 MHz) spectrum of compound **2e** showed eight methyl singlet signals at *δ*_H_ 1.25, 1.24, 1.08, 0.96, 0.88, 0.86, 0.84, 0.81; also showed the hydrogen singlet signal at *δ*_H_ 3.96, which was attached to the α-carbon of proline. The ^13^C-NMR spectrum of compound **2e** showed 35 carbon signals in total. A carbonyl carbon signal was also shown at *δ*_C_ 173.27, indicating the presence of an ester bond. The chemical shifts of C-3 moved to low-field region compared with pyxinol, demonstrating proline was conjugated at C-3 of pyxinol. Based on the above data, compound **2e** was elucidated as (20*S*, 24*R*)-epoxy-3β-*O*-(l-prolyl)-dammarane-12*β*, 25-diol. Compounds **2a**, **2c**, **2d** and **2f** were prepared according to the published literature [31]. The other 20 derivatives are the new compounds.

### 2.3. Cytotoxic Activity Determination on HepG2 Cells

All compounds including lead compounds (pyxinol and ocotillol) and derivatives were evaluated for their cytotoxic activity against the HepG2 cell line by using the CCK-8 assay method with cyclophosphamide (CTX) as the positive control. The results revealed that most derivatives, such as **1a**, **1c**, **1e**, **2a**–**2f**, **3c**, **3e** and **4a**–**4f**, could enhance the anti-tumor activity of lead compounds. In particular, compound **2e** exhibited the highest inhibitory effect on HepG2 cells in vitro with IC_50_ similar to the positive drug (Table 1). Therefore, compound **2e**, the most potent growth inhibitor of the HepG2 cells, was selected for further study.

### 2.4. Effects of Compound 2e on H22 Tumor-Bearing Mice

#### 2.4.1. Effect on Body Weight, Tumor Weight, Tumor Volume and Organ Indexes

Body weights, tumor weights, liver index and kidney index are presented in Table 2. Tumor volume is shown in Figure 3. There is no significant difference in body weights of mice between the groups.

Tumors of the model group reached 10**–**18 mm in diameter on day 14. While CTX, compound **2e** and compound **2e** + CTX treatment resulted in delayed tumor growth compared with model group. It was showed that compound **2e** groups (50, 100 mg/kg) could significantly reduce the tumor weight and volume compared to the model group (*p* < 0.05, *p* < 0.01). Treatment with compound **2e** could decrease the tumor weight and increase the tumor inhibition rate (TIR) in a dose- dependent manner. Both CTX group and compound **2e** + CTX group could significantly reduce tumor weights and volumes compared to the model group (*p* < 0.01). Compound **2e** + CTX group displayed the highest TIR (55.30%).

The data also showed that the liver and kidney indexes were significantly increased in the model group compared with the normal group (*p* < 0.01). While the indexes of compound **2e**-treated mice were basically consistent with the model group. However, the compound **2e** + CTX group, which displayed similar indexes to the normal group, could significantly decreased the liver (*p* < 0.01) and kidney (*p* < 0.05) indexes compared with model group.

#### 2.4.2. Histopathological Examination

Histological analyses of tumors, liver and kidney using H&E staining are shown in Figure 4.

The tumor cells in the model group showed aggressive growth in different forms and sizes. Mitotic activity and different stained tumor cells were also detected. The necrosis in the CTX group was more obvious than in the compound **2e** high-dose group, but had extensive necrosis in the compound **2e** + CTX group.

The liver and kidney cells in the model group and CTX group had many vacuoles and edema. This phenomenon had been alleviated in compound **2e** high-dose group and in compound **2e** + CTX group.

#### 2.4.3. Effects of Compound **2e** Treatment on Cytokine Levels in Mice

The effects of compound **2e** treatment on the expression of IL-2 (Interleukin-2), TNF-*α* (Tumor necrosis factor-*α*) and VEGF (Vascular endothelial growth factor) were listed in Table 3. The level of TNF-*α* and IL-2 were greatly reduced after CTX treatment compared with the normal group (*p* < 0.01), which indicated that the immune system was suppressed by CTX. The level of TNF-*α* was dose-dependently increased in compound **2e** treatment groups compared to the model group, and the highest level was observed in high-dose compound **2e** group. These data indicated that compound **2e** might enhance immune function in H22 tumor-bearing mice by increasing cytokine levels. The levels of IL-2 in compound **2e** groups (moderate and high-dose) were significantly elevated compared to model group (*p* < 0.01), and they were also dose-dependently increased. These results showed that the proliferation and activation of T lymphocytes might be stimulated to cause tumor cell death. Meanwhile, the levels of VEGF expression in compound **2e** treatment groups were all significantly decreased in a dose-dependent manner compared with the model group. CTX and compound **2e** (moderate and high dose) could significantly inhibit the VEGF level in serum (*p* < 0.01). It was indicated that compound **2e** might inhibit tumor growth by effectively reducing tumor angiogenesis.

#### 2.4.4. Effects of Compound **2e** Treatment on Hepatic and Renal Function

The results listed in Table 4 indicated that the serum AST (Aspartate aminotransferase), ALT (Alanine aminotransferase), CRE (Creatinine) and BUN (Blood urea nitrogen) levels dramatically increased in the model group compared with the normal group. The increased levels could be significantly re-regulated with the treatment of compound **2e** (*p* < 0.01) or CTX (*p* < 0.01). The high dosage of compound **2e** showed the best effect. Moreover, the combination group showed the lower values than those in CTX group, which indicated that compound **2e** could increase CTX’s efficiency and reduce its toxicity. These results implied that compound **2e** did less harm to the liver and kidney of tumor-bearing mouse.

### 2.5. Metabolomics Study

#### 2.5.1. Validation of UPLC-QTOF-MS

UPLC-QTOF-MS system was used to perform the metabolomic study. The validation test was first conducted to monitor the durability and stability of system. Serum and liver quality control (QC) samples were run randomly covering the whole analysis process, respectively. The tests included: (1) five consecutive serum or liver QC samples were detected to evaluate the injection precision; (2) five parallel replicates of a serum or liver test sample were assayed to assess the reproducibility; (3) the post-preparation stability of the sample was estimated by detecting one serum or liver test sample that was placed in the autosampler at 10 °C for 0, 2, 8, 12, and 24 h. In the above test, the exact mass/retention time pairs of eight ions in serum or in liver QC samples both in positive and negative modes of ESI (electrospray ionization) were all monitored. The relative standard deviations (RSDs) of peak intensity or retention time were calculated and were listed in Table 5. The validation results indicated the chromatographic and spectrometric system were well-suited for the following metabolomics analysis.

The serum and liver metabolic characteristics of normal group, model group and compound **2e** (100 mg/kg) group in positive ESI (ESI+) and negative ESI (ESI–) modes were acquired. The representative BPI (base peak intensity) chromatograms are shown in Figure 5.

#### 2.5.2. Identification of the Differential Metabolites and Metabolic Pathways

In order to confirm whether the endogenous metabolites were different in serum or in liver between normal, model and compound **2e** groups, principal component analysis (PCA), an unsupervised pattern recognition approach, was firstly performed in both ESI+ and ESI- modes (Figure 6A–D). Both in serum and in liver PCA scores, each spot represented a sample; the tightly clustered QC spots indicated the satisfactory stability of system; three groups were separated indicating their being differential; compound **2e** group was located between the normal group and the model group, indicating that compound **2e** might regulate the metabolic disturbances in H22 tumor-bearing mice.

Aiming at obtaining the maximum separation between the model group and compound **2e** group, orthogonal projections to latent structures discriminant analysis (OPLS-DA), a supervised method of pattern recognition, was then established in both ESI+ and ESI– modes (Figure 6E–H). In the OPLS-DA score plots, each spot also represented a serum or a liver sample. Either in serum samples or in liver samples, the model group and compound **2e** group were both separated with satisfactory R^2^ and Q^2^ parameters.

Permutation test was then performed to validate the prediction ability and the reliability of OPLS-DA. Figure 7A–D showed the test results. All Q^2^-values (blue spots) to the left were lower than the original points to the right, indicating that the OPLS-DA displayed good prediction ability and reliability.

To find the potential biomarkers that contributed to the differentiation between the model group and compound **2e** group, S-plots (Figure 7E–H) under OPLS-DA were generated to visualize the variables. Each spot in S-plots represents an endogenous metabolite in the model group and compound **2e** group. The farther away the spot in the S-plots from the origin, the more significantly the metabolite contribute to the clustering of two groups. The metabolites with VIP (variable importance in the projection) >1.0 and *p* < 0.05 were considered as potential biomarkers. There were 21 robust endogenous metabolites were identified as the candidate biomarkers (listed in Table 6). These biomarkers were marked with measured mass in S-plots. And it was showed that these biomarkers were involved in eight metabolic pathways (Table 7). The relationship between the main metabolites & its involved metabolisms and liver diseases is discussed as follows: 

*Arachidonic acid metabolism* (AM): (1) Arachidonic acid (AA), recognized as the main factor mediating repeated liver injury and compensatory proliferation [62,63], could suppress the growth of hepatic cells and increase cellular transglutaminase 2 transamidase activity. The elevated serum levels of AA had been observed in HCC Patients. (2) 8,9-epoxyeicosatrienoic acid (8,9-EET), a kind of epoxyeicosatrienoic acids (EETs), was closely related to hepatic tumor [64]. (3) 15(S)-hydroperoxy eicosatetraenoic acid (15(S)-HPETE) decreased production of CD31 and VEGF in endothelial cells and had an anti-angiogenic effect in adipose tissue [65,66]. (4) Leukotriene B4 (LTB4) played a role in acute and chronic liver injury. It is important in mediating the inflammatory response, and it is involved in pathogenesis of several autoimmune, inflammatory diseases as well as in tumor proliferation [67]. (5) Prostaglandin I2 (PGI2), an important downstream from AA, is also a mediator of tumor progression [68]. Exogenous PGI2 could decrease tumor cell proliferation. In this experiment, the elevated levels (AA, 8, 9-EET, 5-HPETE, LTB4) and decreased levels (15(S)-HPETE, PGI2) in the model group indicated the imbalance of arachidonic acid metabolism. While the above levels could be re-regulated by compound **2e** treatment.

*Linoleic acid metabolism* (LM): (1) Phosphatidylcholine (16:0/16:1) (PC(16:0/16:0)), a precursor of lipid inflammatory mediators, was specifically localized in colorectal cancer region [69,70]. (2) 12,13-epoxyoctadecenoic acid (12,13-EpOME), a metabolite of CYP monooxygenase, has proinflammatory effect. In this experiment, the elevated levels of PC (16:0/16:1) and 12, 13-EpOME observed in the model group, could be re-regulated by compound **2e**.

*Sphingolipid metabolism* (SM): Sphinganine 1-phosphate (Sa1-P) and sphingosine 1- phosphate (So1-P) were the bioactive phospholipids involved in proliferation of HepG2 cells in vivo [71,72]. In the current study, Sa1-P and So1-P increased in the model group, could be re-regulated by compound **2e.**

*Retinol metabolism* (RM): In liver tumors, retinyl ester levels were significantly decreased [73]. Hydroxyretinoic acid and all-trans-5, 6-Epoxyretinoic acid could inhibit the growth of breast cancer cell and rat rhabdomyosarcoma cell [74]. All-trans-retinoic acid, the active metabolite of vitamin A, could affect liver cancer stem cells. All the above metabolites decreased in the model group and could be up-regulated following compound **2e** treatment.

*Porphyrin and chlorophyll metabolism* (PCM): The level of bilirubin was significantly elevated in the H22 liver tumor model group [75]. In our study, bilirubin was also markedly decreased in the model group, and was re-regulated following compound **2e** treatment.

*Tryptophan metabolism* (TryM): l-tryptophan, an essential amino acid for human, could prevent hepatic fibrosis progression [76]. It was shown that l-tryptophan decreased in the model group and could be re-regulated by compound **2e**.

*Glycerophospholipid metabolism* (GlyM): Altered glycerophospholipid metabolism (Lyso-PC a C18:1) has been reported in both blood and liver tissue samples from nonalcoholic fatty liver disease patients [77]. It had also been disturbed in the liver injury mouse model [78]. In our study, LysoPC (18:1(9Z)/0:0) increased in the model group and could be re-regulated by compound **2e**.

*α-Linolenic acid metabolism* (ALAM): Alterations in phospholipid and fatty acid metabolism may play key roles in hepatocarcinogenesis. It was reported that the tumor exhibited significantly lower level of α-linolenic acid than peritumoral liver tissue [79]. Stearidonic acid inhibited intestinal tumor development and cancer cell proliferation, and could be potentially chemo-preventive [80]. All above metabolites, decreased in the model group and could be re-regulated by compound **2e**.

In addition, the predictive ROC curves, generated by using 21 candidate biomarkers, showed that the candidate biomarkers were potential diagnostic markers for liver tumor (Figure 8A) and were contributed to compound **2e** treatment (Figure 8B). The area under curve (AUC) values and *p*-values of the biomarkers in ROC curves were listed in Table 8.

In order to characterize and visualize the biomarkers’ relative abundance in three groups, the heatmap (Figure 9A) was then generated with green color representing low abundance and red color representing high abundance. Moreover, the established metabolic network of the biomarkers was shown in Figure 9B.

## 3. Materials and Method

### 3.1. Chemistry

The ^1^H-NMR and ^13^C-NMR spectra using TMS as internal standard was assayed in CDCl_3_ or Pyridin-d_5_ on Bruker AV-600 spectrometer (Bruker Co., Karlsruhe, Germany). Chemical shifts were expressed in *δ* values (ppm). High-resolution electrospray ionization mass spectrometry (HR-ESI-MS) was performed on Waters Xevo G2-XS QTOF mass spectrometer (Waters Co., Milford, MA, USA). Chemical reagents and solvents were purchased from Saen Chemical Technology Co., Ltd. (Shanghai, China).

#### 3.1.1. The Synthesis of Compounds **1a**–**1f** and **3a**–**3f**

At room temperature, EDC (3.15 mmol) and DMAP (1.05 mmol) in dry CH_2_Cl_2_ (20 mL) were added to *N*-Boc protected amino acids (4.20 mmol) solution in a triangle flask with a magnetic stirrer. Pyxinol (2.10 mmol) was added 30 min later. The reaction mixtures were shaken for 12 h and progress was monitored by TLC (*n*-hexane/ethyl acetate 3:1). Then the organic solution was washed with saturated aqueous NaHCO_3_ solution (3 × 10 mL), water (3 × 10 mL) and brine (3 × 10 mL), dried (MgSO_4_), filtered and concentrated under reduced pressure to give the crude product. The crude products were chromatographed using silica gel and eluted with *n*-hexane and ethyl acetate (5:1) to obtain the pure products **1a**–**1f** used as substrates for the Boc deprotection reaction. The ocotillol N-Boc protected amino acids derivatives **3a**–**3f** using the same procedure described above for compounds **1a***–***1f**, except all reactants were dissolved in THF.

(20*S*, 24*R*)-Epoxy-3β-*O*-(Boc-l-alanyl)-dammarane-12β, 25-diol (1a) 

Light yellow powder (yield 96.8%), ^1^H-NMR (CDCl_3_, 600 MHz) *δ* 5.55 (s, 1H), 5.04 (s, 1H), 4.49 (dd, *J* = 9.3, 7.1 Hz, 1H), 4.30–4.26 (m, 1H), 3.84–3.81 (m, 1H), 3.51–3.46(m, 1H), 2.19–2.14 (m, 1H), 2.06–1.92 (m, 2H), 1.89–1.81 (m, 3H),1.72–1.62 (m, 6H), 1.57–1.50 (m, 4H), 1.42 (s, 12H, 4×-CH_3_), 1.37 (d, *J* = 7.2 Hz, 3H), 1.29–1.27 (m, 2H), 1.25 (s, -CH_3_), 1.24 (s, -CH_3_), 1.12-1.09 (m, 2H), 1.07 (s, -CH_3_), 0.96 (s, -CH_3_), 0.94–0.92 (m, 1H), 0.87 (s, -CH_3_), 0.86 (s, -CH_3_), 0.84 (s, -CH_3_), 0.81 (s, -CH3). ^13^C-NMR (CDCl_3_, 150 MHz) *δ* 173.24, 155.19, 86.73, 85.65, 82.07, 79.88, 71.15, 70.31, 56.26, 52.23, 50.64, 49.77, 49.62, 48.18, 39.99, 38.80, 38.26, 37.29, 34.97, 32.84, 31.57, 31.43, 30.79, 28.81, 28.58 (×3C), 28.16, 27.82, 26.37, 25.22, 23.89, 19.41, 19.19, 18.38, 16.67, 16.59, 15.62.HR-MS (ESI, m/z) [M + H]^+^ calcd. for C_38_H_65_NO_7_, 648.4839, found: 648.4834.

(20*S*, 24*R*)-Epoxy-3β-*O*-(Boc-l-methinyl)-dammarane-12β, 25-diol (**1b**)

Yellow powder (yield 86.2%), ^1^H-NMR (CDCl_3_, 600 MHz) *δ* 5.07 (s, 1H), 4.51–4.48 (m, 1H), 4.39–4.37 (m, 1H), 3.84–3.81 (m, 1H), 3.49 (td, *J =* 10.5, 4.5 Hz, 1H), 2.54–2.50 (m, 2H), 2.19–2.15 (m, 2H), 2.07 (s, -SCH_3_), 2.02-1.81 (m, 7H), 1.71–1.50 (m, 10H), 1.42 (s, 3×-CH_3_), 1.29–1.27 (m, 2H), 1.25 (s, -CH_3_), 1.24 (s, -CH_3_), 1.12-1.09 (m, 2H), 1.07 (s, -CH_3_), 1.05–1.00 (m, 1H), 0.96 (s, -CH_3_), 0.87 (s, -CH_3_), 0.86 (s, -CH_3_), 0.84 (s, -CH_3_), 0.82 (s, -CH_3_). ^13^C-NMR (CDCl_3_, 150 MHz) *δ* 172.18, 155.54, 86.72, 85.64, 82.57, 80.11, 71.13, 70.31, 56.27, 53.46, 52.22, 50.63, 49.61, 48.18, 39.98, 38.78, 38.18, 37.28, 34.96, 32.83, 32.74, 31.56, 31.42, 30.23, 28.80, 28.55 (×3C), 28.32, 28.13, 27.81, 26.36, 25.21, 23.89, 18.36 (×2C), 16.75, 16.57, 15.69, 15.61. HR-MS (ESI, m/z) [M + H]^+^ calcd. for C_40_H_69_NO_7_S, 708.4873, found: 708.4876.

(20*S*, 24*R*)-Epoxy-3β-*O*-(Boc-l-glycyl)-dammarane-12β, 25-diol (**1c**) 

White powder (yield 83.7%), ^1^H-NMR (CDCl_3_, 600 MHz) *δ* 4.97 (s, 1H), 4.53–4.50 (m, 1H), 3.87 (d, *J =* 5.1 Hz, 2H), 3.84–3.81 (m, 1H), 3.51–3.46 (td, *J* = 10.5, 4.6 Hz, 1H), 2.19–2.14 (m, 1H), 2.05–1.81 (m, 6H), 1.72–1.46 (m, 9H), 1.43 (s, 3×-CH_3_), 1.41–1.37 (m, 1H), 1.30–1.27 (m, 2H), 1.25 (s, -CH_3_), 1.24 (s, -CH_3_), 1.12-1.09 (m, 2H), 1.07 (s, -CH_3_), 1.05-1.00 (m, 1H), 0.96 (s, -CH_3_), 0.87 (s, -CH_3_), 0.85 (s, -CH_3_), 0.82 (s, 2×-CH_3_). ^13^C-NMR (CDCl_3_, 150 MHz) *δ* 170.33, 155.84, 86.73, 85.65, 82.36, 80.08, 71.15, 70.32, 56.27, 52.23, 50.64, 49.61, 48.19, 42.87, 39.99, 38.82, 38.20, 37.28, 34.97, 32.83, 31.56, 31.43, 28.81, 28.55 (×3C), 28.21, 28.14, 27.82, 26.36, 25.22, 23.89, 18.37 (×2C), 16.60 (×2C), 15.62. HR-MS (ESI, m/z) [M + H]^+^ calcd. for C_37_H_63_NO_7_, 634.4683, found: 634.4687.

(20*S*, 24*R*)-Epoxy-3β-*O*-(Boc-l-leucyl)-dammarane-12β,25-diol (**1d**) 

Light yellow powder (yield 79.1%), ^1^H-NMR (CDCl_3_, 600 MHz) *δ* 4.83 (s, 1H), 4.49–4.46 (m, 1H), 4.28–4.23 (m, 1H), 3.84–3.81 (m, 1H), 3.49 (td, *J* = 10.5, 4.5 Hz, 1H), 2.19–2.14 (m, 1H), 2.02–1.98 (m, 2H), 1.87–1.84 (m, 3H), 1.74–1.43 (m, 14H), 1.41 (s, -CH_3_), 1.30–1.27 (m, 2H), 1.25 (s, -CH_3_), 1.24 (s, -CH_3_), 1.12–1.09 (m, 2H), 1.07 (s, -CH_3_), 1.05–1.00 (m, 1H), 0.96 (s, -CH_3_), 0.93 (d, *J* = 1.6 Hz, 2× -CH_3_), 0.87 (s, -CH_3_), 0.86 (s, -CH_3_), 0.84 (s, -CH_3_), 0.82 (s, -CH_3_). ^13^C-NMR (CDCl_3_, 150 MHz) *δ* 73.33, 155.58, 86.73, 85.65, 82.02, 79.86, 71.16, 70.31, 56.30, 52.77, 52.23, 50.64, 49.62, 48.18, 42.30, 39.99, 38.80, 38.18, 37.28, 34.98, 32.84, 31.56, 31.42, 29.92, 28.81, 28.57 (×3C), 28.25, 28.13, 27.82, 26.36, 25.22, 25.07, 23.88, 23.17, 22.13, 18.38, 16.71, 16.57, 15.62. HR-MS (ESI, m/z) [M + H]^+^ calcd. for C_41_H_71_NO_7_, 690.5309, found: 690.5314.

(20*S*, 24*R*)-Epoxy-3β-*O*-(Boc-l-prolyl)-dammarane-12β, 25-diol (**1e**)

Yellow powder (yield 86.2%), ^1^H-NMR (CDCl_3_, 600 MHz) *δ* 4.53–4.49 (m, 1H), 4.35–4.33 (m, 1H), 4.27–4.25 (m, 1H), 3.88–3.85 (m, 1H), 3.56–3.48 (m, 2H), 2.27–2.16 (m, 2H), 2.08–1.86 (m, 10H), 1.71–1.64 (m, 6H), 1.61–1.55 (m,3H), 1.47 (s, -CH_3_), 1.44 (s, 3×-CH_3_), 1.33–1.30 (m, 2H), 1.30 (s, -CH_3_), 1.29 (s, -CH_3_), 1.16–1.13 (m, 1H), 1.12 (s, -CH_3_), 1.10–1.04 (m, 1H), 1.00 (s, -CH_3_), 0.98–0.93 (m, 1H), 0.92 (s, -CH_3_), 0.90 (s, -CH_3_), 0.87 (s, -CH_3_). ^13^C-NMR (CDCl_3_, 150 MHz) *δ* 172.98, 154.09, 86.73, 85.65, 81.41, 71.15, 70.30, 59.66, 56.28, 53.84, 52.23, 50.63, 49.63, 48.19, 46.65, 46.47, 39.99, 38.79, 38.22, 37.28, 34.97, 32.84, 31.56, 31.44, 30.40, 28.81, 28.72, 28.63, 28.14, 27.84, 26.37, 25.23, 24.51, 23.96, 23.69, 18.87, 18.38, 17.62, 16.59, 15.62. HR-MS (ESI, m/z) [M + H]^+^ calcd. for C_40_H_67_NO_7_, 674.4996, found: 674.5001.

(20S, 24R)-Epoxy-3β-O-(Boc-l-valyl)-dammarane-12β, 25-diol (**1f**)

Light yellow powder (yield 78.6%), ^1^H-NMR (CDCl_3_, 600 MHz) δ 4.95 (s, 1H), 4.50–4.47 (m, 1H), 4.20–4.18 (m, 1H), 3.84–3.81 (m, 1H), 3.51–3.46 (td, *J* =5, 4.6 Hz, 1H), 2.19–2.14 (m, 2H), 2.05–1.81 (m, 6H), 1.70–1.46 (m, 9H), 1.42 (s, 4×-CH_3_), 1.32–1.27 (m, 2H), 1.09 (m, 2H), 1.25 (s, -CH_3_), 1.24 (s, -CH_3_), 1.12–1.09 (m, 1H), 1.07 (s, -CH_3_), 1.04–1.01 (m, 1H), 0.97 (s, -CH_3_), 0.96 (s, -CH_3_), 0.95 (s, -CH_3_), 0.87 (s, -CH_3_), 0.86 (s, -CH_3_), 0.84 (s, -CH_3_), 0.83 (s, -CH_3_). ^13^C-NMR (CDCl_3_, 150 MHz) δ 172.27, 155.96, 86.73, 85.65, 82.26, 79.83, 71.15, 70.31, 59.11, 56.29, 52.23, 50.64, 49.61, 48.18, 39.99, 38.80, 38.10, 37.27, 34.98, 32.83, 31.56, 31.42, 28.81, 28.57 (×3C), 28.25, 28.13, 27.81, 26.36, 25.22, 23.98, 19.54, 18.37 (×3C), 17.43, 16.77, 16.57, 15.62. HR-MS (ESI, m/z) [M + H]^+^ calcd. for C_40_H_69_NO_7_,676.5152, found: 676.5156.

(20S, 24*R*)-Epoxy-3β-*O*-(Boc-l-alanyl)-dammarane-6α,12β,25-triol (**3a**)

White powder (yield 56.8%), ^1^H-NMR (CDCl_3_, 600 MHz) *δ* 5.35 (td, *J* = 10.5, 4.5 Hz, 1H), 4.92 (s, 1H), 4.46–4.43 (m, 1H), 3.84–3.81 (dd, *J =* 8.7, 6.8 Hz, 1H), 3.52–3.47 (m, 1H), 3.36 (s, 1H), 2.51–2.45 (m, 4H), 2.19–2.15 (m, 1H), 2.02–1.81 (m, 4H), 1.65–1.50 (m, 6H), 1.41 (s, 4×-CH_3_), 1.32–1.27 (m, 2H), 1.25 (s, -CH_3_), 1.24 (s, -CH_3_), 1.14–1.11 (m, 2H), 1.10 (s, -CH_3_), 1.07 (s, -CH_3_), 1.04–1.03 (m, 1H), 0.97 (s, -CH_3_), 0.96 (s, -CH_3_), 0.90 (s, -CH_3_), 0.88 (s, -CH_3_). ^13^C-NMR (CDCl_3_, 150 MHz) *δ* 172.33, 155.90, 86.60, 85.60, 80.71, 79.51, 71.09, 70.81, 70.27, 61.48, 58.97, 51.98, 50.02, 49.14, 47.99, 42.62, 40.86, 39.41, 38.35, 37.87, 35.03, 32.75, 31.34, 30.59, 28.70, 28.54 (×3C), 28.04, 27.66, 26.29, 25.10, 23.42, 18.22, 17.62, 17.51, 16.91, 16.75.HR-MS (ESI, m/z) [M + H]^+^ calcd. for C_38_H_65_NO_8_, 664.4788, found: 664.4785.

(20*S*, 24*R*)-Epoxy-3β-*O*-(Boc-l-methinyl)-dammarane-6α,12β,25-triol (3**b**)

Yellow powder (yield 63.4%), ^1^H-NMR (CDCl_3,_ 600 MHz) *δ* 5.59 (s, 1H), 5.37–5.29 (m, 1H), 5.06 (s, 1H), 4.50–4.47 (m, 1H), 4.41–4.32 (m, 1H), 3.84–3.81 (m, 1H), 3.51–3.46 (m, 1H), 2.54–2.51 (m, 2H), 2.31–2.27 (m, 2H), 2.12–2.09 (m, 2H), 2.07 (s, -SCH_3_), 1.72–1.62 (m, 5H), 1.52–1.48 (m, 5H), 1.42 (s, 3×-CH_3_), 1.31–1.29 (m, 5H), 1.24 (d, *J =* 2.8 Hz, 2×-CH_3_), 1.24 (s, -CH_3_), 1.09 (s, -CH_3_), 1.07 (s, -CH_3_), 1.04–1.00 (m, 2H), 0.99 (s, -CH_3_), 0.97 (s, -CH_3_), 0.95-0.93 (m, 1H), 0.90 (d, *J =* 4.2 Hz, 2×-CH_3_). ^13^C-NMR (CDCl_3_, 150 MHz) *δ* 172.19, 155.49, 86.60, 85.58, 81.87, 80.12, 70.82, 70.57, 70.28, 58.94, 53.37, 51.97, 50.01, 49.11, 47.98, 42.57, 40.77, 39.32, 38.29, 34.12, 32.73, 32.51, 31.33, 30.58, 30.18, 28.70, 28.46 (×3C), 28.03, 27.66, 26.26, 25.10, 23.37, 18.20, 17.49, 16.97, 16.73, 15.62. HR-MS (ESI, m/z) [M + H]^+^ calcd. for C_40_H_69_NO_8_S,724.4822, found:724.4827.

(20*S*, 24*R*)-Epoxy-3β-*O*-(Boc-l-glycyl)-dammarane-6α,12β,25-triol (**3c**)

White powder (yield 51.6%), ^1^H-NMR (CDCl_3_, 600 MHz) *δ* 5.60 (s, 1H), 5.00 (s, 1H), 4.52–4.49 (m, 1H), 4.12–4.09 (m, 1H), 3.91–3.90 (m, 2H), 3.85–3.83 (m, 1H), 3.53–3.48 (m, 1H), 2.21–2.17 (m, 1H), 2.07–1.96 (m, 3H), 1.93–1.80 (m, 4H), 1.72–1.65 (m, 5H), 1.62 (s, -CH_3_), 1.58–1.49 (m, 3H), 1.45 (s, 3×-CH_3_), 1.33–1.29 (m, 2H), 1.27 (s, -CH_3_), 1.26 (s, -CH_3_), 1.16 (s, -CH_3_), 1.09 (s, -CH_3_), 1.05 (s, -CH_3_), 0.98–0.97 (m, 2H), 0.93 (s, -CH_3_), 0.92 (s, -CH_3_). ^13^C-NMR (CDCl_3_, 150 MHz) *δ* 170.20, 155.64, 86.48, 85.44, 81.95, 79.93, 70.79, 70.14, 68.35, 61.30, 51.81, 49.89, 49.0, 47.88, 47.05, 42.67, 40.88 38.94, 38.33, 38.20, 32.60, 31.24, 31.19, 30.68, 28.58, 28.33 (×3C), 27.89, 27.53, 26.13, 24.95, 23.34, 18.10, 17.48, 16.93, 16.49. HR-MS (ESI, m/z) [M + H]^+^ calcd. for C_37_H_63_NO_8_, 650.4632, found: 650.4631.

(20*S*, 24*R*)-Epoxy-3β-*O*-(Boc-l-leucyl)-dammarane-6α,12β,25-triol (**3d**)

Light yellow powder (yield 43.8%), ^1^H-NMR (CDCl_3_, 600 MHz) *δ* 5.69 (s, 1H), 5.25 (s, 1H), 4.89 (s, 1H,), 4.40–4.36 (m, 1H), 4.06 (d, *J =* 11.1 Hz, 1H), 3.88–3.85 (m, 1H), 3.55–3.49 (m, 1H), 3.22–3.20 (m, 1H), 2.23–2.19 (m, 1H), 2.15 (m, 3H), 1.94–1.83 (m, 6H), 1.71–1.59 (m, 9H), 1.46 (s, 3×-CH_3_), 1.36 (s, -CH_3_), 1.35 (s, -CH_3_), 1.33-1.31 (m, 1H), 1.29 (s, -CH_3_), 1.18 (s, -CH_3_), 1.11 (s, -CH_3_), 1.09 (s, -CH_3_), 1.05-1.03 (m, 3H), 0.97 (s, -CH_3_), 0.96 (s, -CH_3_), 0.92 (s, -CH_3_), 0.90 (s, -CH_3_). ^13^C-NMR (CDCl_3_, 150 MHz) *δ* 172.79, 155.75, 86.62, 85.57, 81.76, 78.80, 77.37, 72.41, 70.34, 68.53, 61.50, 56.09, 52.29, 50.39, 49.14, 48.63, 47.19, 42.14, 39.95, 39.15, 38.44, 37.27, 32.74, 31.93, 31.60, 30.90, 28.72, 28.52 (×3C), 28.15, 27.56, 26.87, 26.27, 25.09, 24.91, 22.98, 22.20, 18.18, 17.57, 17.08, 16.40. HR-MS (ESI, m/z) [M + H]^+^ calcd. for C_41_H_71_NO_8_, 706.5258, found:706.5255.

(20*S*, 24*R*)-Epoxy-3β-*O*-(Boc-l-prolyl)-dammarane-6α,12β,25-triol (**3e**).

Light yellow powder (yield 69.1%), ^1^H-NMR (CDCl_3_, 600 MHz) *δ* 5.32 (s, 1H), 4.53–4.48 (m, 1H), 4.22 (m, 1H), 4.27–4.26 (m, 1H), 4.20–4.09 (m, 1H), 3.88–3.86 (m, 1H), 3.54–3.48 (m, 1H), 2.23–2.19 (m, 2H), 2.05–1.86 (m, 10H), 1.76–1.55 (m, 10H), 1.46 (s, -CH_3_), 1.44 (s, 3×-CH_3_), 1.36–1.34 (m, 2H), 1.29 (s, 2×-CH_3_), 1.19-1.05 (m, 1H), 1.11 (s, -CH_3_), 1.09 (s, -CH_3_), 1.02 (s, -CH_3_), 0.93 (s, -CH_3_), 0.90 (s, -CH_3_). ^13^C-NMR (CDCl_3_, 150 MHz) *δ* 172.98, 153.98, 86.60, 85.60, 80.85, 80.08, 72.33, 70.79, 70.28, 59.53, 58.93, 52.00, 50.16, 49.13, 47.95, 46.61, 46.41, 42.77, 41.04, 39.56, 38.27, 32.77, 31.35, 30.58, 30.29, 29.82, 28.74, 28.60, 28.04 (×3C), 27.68, 26.28, 25.11, 24.48, 23.61, 18.25, 17.59, 17.41, 16.75. HR-MS (ESI, m/z) [M + H]^+^ calcd. for C_40_H_67_NO_8_, 690.4945, found:690.4941.

(20*S*, 24*R*)-Epoxy-3β-*O*-(Boc-l-valyl)-dammarane-6α,12β,25-triol (**3f**)

White powder (yield 52.3%), ^1^H-NMR (CDCl_3_, 600 MHz) *δ* 5.38-5.34 (td, *J =* 10.5, 4.5 Hz, 1H), 4.95 (s, 1H), 4.48–4.45 (m, 1H), 4.21–4.18 (m, 1H), 3.84–3.81 (m, 1H), 3.51–3.47 (m, 1H), 2.18–2.14 (m, 2H), 2.02–1.79 (m, 6H), 1.73–1.48 (m, 10H), 1.41 (s, 3×-CH_3_), 1.40 (s, 2×-CH_3_), 1.32-1.29 (m, 2H), 1.25 (s, -CH_3_), 1.24 (s, -CH_3_), 1.10 (s, -CH_3_), 1.07 (s, -CH_3_), 1.02 (s, -CH_3_), 0.98 (s, -CH_3_), 0.95 (s, -CH_3_), 0.93-0.91 (m, 1H), 0.89 (s, -CH_3_). ^13^C-NMR (CDCl_3_, 150 MHz) *δ* 172.28, 155.87, 86.60, 85.60, 81.78, 79.83, 72.77, 72.20, 70.79, 70.28, 58.93, 51.97, 49.14, 47.97, 42.63, 40.99, 39.57, 38.34, 37.78, 32.73, 31.33 (×2C), 31.22, 30.88, 30.60, 28.69, 28.47 (×3C), 28.03, 27.65, 26.28, 25.09, 23.35, 19.70, 19.51, 18.20, 17.45, 16.94, 16.71. HR-MS (ESI, m/z) [M + H]^+^ calcd. for C_40_H_69_NO_8_, 692.5101, found: 692.5104.

#### 3.1.2. Synthesis of Compounds **2a**–**2f** and **4a**–**4f**

At room temperature, the intermediate of *N*-Boc-amino acids conjugated pyxinol was directly dissolved in dry CH_2_Cl_2_ (5 mL), TFA (2 mL) was then added gently, and the mixture was stirred for 2 h. The reaction solution was washed with NaHCO_3_, water, and saturated brine, dried (MgSO_4_), filtered, and concentrated. The concentrated residue was purified by silica gel to afford corresponding conjugate **2a**–**2f**, dichloromethane and methanol (60:1-15:1) were used as eluting solvents. The **4a**−**4f** were obtained by the same procedure described above, except the volume of eluting solvents dichloromethane and methanol are 30:1-5:1.

(20*S*, 24*R*)-Epoxy-3*β*-*O*-(l-alanyl)-dammarane-12*β*, 25-diol (**2a**)

Light yellow powder (yield 91.8%), ^1^H-NMR (Pyridin-d_5_, 600 MHz) *δ* 5.83 (s, 1H), 4.89 (s, 1H), 4.74 (dd, *J =* 10.9, 5.7 Hz, 1H), 4.03–3.99 (m, 1H), 3.82–3.73 (m, 2H), 2.32–2.18 (m, 2H), 2.03–1.84 (m, 5H), 1.69–1.57 (m, 5H), 1.54 (s, -CH_3_), 1.50 (s, -CH_3_), 1.49–1.47 (m, 2H), 1.45–1.35 (m, 4H), 1.34 (s, -CH_3_), 1.32 (s, -CH_3_), 1.29–1.27 (m, 2H), 1.12–1.08 (m, 1H), 1.03 (s, -CH_3_), 0.95 (s, -CH_3_), 0.94 (s, -CH_3_), 0.93 (s, -CH_3_), 0.90-0.86 (m, 1H), 0.81 (s, -CH_3_). ^13^C-NMR (Pyridin-d_5_, 150 MHz) *δ* 177.01, 87.09, 86.03, 81.10, 71.41, 70.69, 56.45, 52.53, 51.41, 50.96, 50.15, 48.75, 40.35, 38.99, 38.61, 37.52, 35.34, 33.19, 32.79, 32.07, 29.15, 28.44, 28.08, 27.64, 27.33, 25.89, 24.31, 21.71, 18.78, 18.70, 17.09, 16.82, 15.89. HR-MS (ESI, m/z) [M+H]^+^ calcd. for C_33_H_57_NO_5_, 548.4315, found: 548.4317.

(20S, 24R)-Epoxy-3*β*-*O*-(l-methinyl)-dammarane-12*β*, 25-diol (**2b**).

Yellow powder (yield 82.3%), ^1^H-NMR (CDCl_3_, 600 MHz) *δ* 5.55 (s, 1H), 4.51–4.49 (m, 1H), 3.83–3.81 (m, 1H), 3.60–3.58 (m, 1H), 3.51–3.46 (m, 1H), 2.64–2.62 (m, 2H), 2.18–2.14 (m, 1H), 2.08 (s, -CH_3_), 2.07–1.93 (m, 5H), 1.86–1.83 (m, 4H), 1.71–1.69 (m, 1H), 1.66–1.60 (m, 4H), 1.56–1.50 (m, 2H), 1.47–1.42 (m,3H), 1.29–1.27 (m, 2H), 1.25 (s, -CH_3_), 1.24 (s, -CH_3_), 111–1.09 (m,1H), 1.07 (s, -CH_3_), 1.05–1.02 (m,1H), 0.96 (s, -CH_3_), 0.88 (s, -CH_3_), 0.86 (s, -CH_3_), 0.84 (s, -CH_3_), 0.82 (s, -CH_3_). ^13^C-NMR (CDCl_3_, 150 MHz) *δ* 175.47, 86.93, 85.85, 82.20, 71.35, 70.52, 56.48, 54.04, 52.44, 50.85, 49.83, 48.39, 40.26, 39.02, 38.43, 37.50, 35.17, 34.11, 33.04, 31.78, 31.63, 30.93, 30.12, 28.56, 28.34, 28.02, 26.57, 25.43, 24.14, 18.59, 18.57, 17.00, 16.79, 15.83, 15.75. HR-MS (ESI, m/z) [M + H]^+^ calcd. for C_35_H_61_NO_5_, 608.4348, found: 608.4350.

(20*S*, 24*R*)-Epoxy-3*β*-*O*-(l-glycyl)-dammarane-12*β*, 25-diol (**2c**)

White powder (yield 81.9%), ^1^H-NMR (Pyridin-d_5_, 600 MHz) *δ* 5.81 (s, 1H), 4.87 (s, 1H), 4.73 (dd, *J =* 11.5, 4.9 Hz, 1H), 4.00–3.97 (m, 1H), 3.92–3.87 (m, 1H), 3.76–3.66 (m, 2H), 2.29–2.15 (m, 2H), 2.00–1.81 (m, 5H), 1.69–1.52 (m, 5H), 1.51 (s, -CH_3_), 1.48–1.33 (m, 5H), 1.31 (s, -CH_3_), 1.29 (s, -CH_3_), 1.27–1.24 (m,2H), 1.09–1.04 (m, 2H), 1.00 (s, -CH_3_), 0.93 (s, -CH_3_), 0.89 (s, -CH_3_), 0.88 (s, -CH_3_),0.85–0.82(m, 1H), 0.79 (s, -CH_3_).^13^C-NMR (Pyridin-d5, 150 MHz) *δ* 174.94, 87.08, 86.02, 81.31, 71.41, 70.69, 56.48, 52.53, 50.97, 50.14, 48.75, 45.19, 40.35, 39.02, 38.51, 37.51, 35.34, 33.18, 32.79, 32.07, 29.15, 28.39, 28.07, 27.63, 27.33, 25.89, 24.35, 18.80, 18.70, 17.02, 16.84, 15.88. HR-MS (ESI, m/z) [M + H]^+^ calcd. for C_32_H_55_NO_5_, 534.4158, found: 534.4155.

(20*S*, 24*R*)-Epoxy-3*β-O*-(l-leucyl)-dammarane-12*β*, 25-diol (**2d**) 

Light yellow powder (yield 73.1%), ^1^H-NMR (CDCl_3_, 600 MHz) *δ* 5.58 (s, 1H), 4.52–4.49 (m, 1H), 3.85-3.83 (m, 1H), 3.53–3.51 (m, 1H), 3.51–3.46 (m, 1H), 2.21–2.17 (m, 1H), 2.07 –1.77 (m, 10H), 1.73–1.1.64 (m, 5H), 1.58–1.53 (m, 3H), 1.49–1.44 (m, 3H), 1.31–1.29 (m, 2H), 1.27 (s, -CH_3_), 1.26 (s, -CH_3_), 1.13–1.11 (m, 1H), 1.09 (s, -CH_3_), 0.98 (s, -CH_3_), 0.94 (dd, *J =* 9.6, 6.6, Hz, 2×-CH_3_), 0.90 (s, -CH_3_), 0.88 (s, -CH_3_), 0.86 (s, CH_3_), 0.85 (s, -CH_3_). ^13^C-NMR (CDCl_3_, 150 MHz) *δ* 176.06, 86.65, 85.56, 81.57, 71.09, 70.25, 56.23, 53.35, 52.17, 50.57, 49.54, 48.11, 44.05, 39.92, 38.74, 38.14, 37.23, 34.91, 32.75, 31.50, 31.34, 28.74, 28.22, 28.06, 27.73, 26.29, 25.15, 25.00, 23.84, 23.15, 21.99, 18.32, 18.28, 16.68, 16.51, 15.54. HR-MS (ESI, m/z) [M + H]^+^ calcd. for C_36_H_63_NO_5_, 590.4784, found: 590.4789.

(20*S*, 24*R*)-Epoxy-3*β*-*O*-(l-prolyl)-dammarane-12*β*, 25-diol (**2e**)

Light yellow powder (yield 83.5%), ^1^H-NMR (CDCl_3_, 600 MHz) *δ* 5.58 (s, 1H), 4.54-451 (m, 1H), 3.96 (s, 1H), 3.84–3.81 (m, 1H), 3.49 (td, *J =* 10.5, 4.6 Hz, 1H), 3.19–3.07 (m, 2H), 2.26–2.15 (m, 2H), 2.01–1.76 (m, 8H), 1.71–1.60 (m, 5H), 1.56–1.39 (m, 6H), 1.29–1.27 (m, 2H), 1.25 (s, -CH_3_), 1.24 (s, -CH_3_), 1.11–1.09 (m, 1H), 1.08 (s, -CH_3_), 1.05-1.01 (m, 2H), 0.96 (s, -CH_3_), 0.93–0.90 (m, 1H), 0.88 (s, -CH_3_), 0.86 (s, -CH_3_), 0.84 (s, -CH_3_), 0.81 (s, -CH_3_). ^13^C-NMR (CDCl_3_, 150 MHz) *δ* 173.27, 86.49, 85.38, 82.28, 70.92, 70.15, 59.87, 56.03, 52.02, 50.43, 49.36, 47.93, 46.65, 39.78, 38.57, 38.10, 37.09, 34.74, 32.63, 31.39, 31.19, 30.11, 28.59, 28.08, 27.96, 27.54, 26.02, 25.02, 24.97, 23.70, 18.19, 18.15, 16.48, 16.37, 15.40. HR-MS (ESI, m/z) [M + H]^+^ calcd. for C_35_H_59_NO_5_, 574.4471, found: 574.4469.

(20*S*, 24*R*)-Epoxy-3*β*-*O*-(l-valyl)-dammarane-12*β*, 25-diol (**2f**)

Light yellow powder (yield 72.7%), ^1^H-NMR (Pyridin-d_5_, 600 MHz) *δ* 5.82 (s, 1H), 4.87 (s, 1H), 4.72 (dd, *J =* 11.6, 4.8 Hz, 1H), 4.01–3.98 (m, 1H), 3.77–3.71 (td, *J =* 10.4, 4.6 Hz, 1H), 3.52 (d, *J =* 4.9 Hz, 1H), 3.30–2.16 (m, 3H), 2.03–1.82 (m, 5H), 1.73–1.54 (m, 6H), 1.52 (s, -CH3), 1.47–1.34 (m, 5H), 1.32 (s, -CH_3_), 1.30 (s, -CH_3_), 1.28–1.26 (m, 2H), 1.12 (d, *J =* 6.8 Hz, -CH_3_), 1.07 (d, *J =* 6.8 Hz -CH_3_), 1.02 (s, -CH_3_), 0.97 (s, -CH_3_), 0.94 (s, 2×-CH_3_), 0.92–0.85 (m, 2H), 0.82 (s, -CH_3_). ^13^C-NMR (Pyridin-d5, 150 MHz) *δ* 176.02, 87.10, 86.04, 81.31, 71.42, 70.70, 61.38, 56.50, 52.55, 50.97, 50.16, 48.76, 40.37, 39.01, 38.50, 37.53, 35.37, 33.20, 32.93, 32.80, 32.08, 30.40, 28.53, 28.09, 27.65, 27.34, 25.90, 24.41, 20.33, 19.64, 18.71, 17.67, 17.23, 16.83, 15.91. HR-MS (ESI, m/z) [M + H]^+^ calcd. for C_35_H_61_NO_5_, 576.4628, found: 576.623.

(20*S*, 24*R*)-Epoxy-3*β*-*O*-(l-alanyl)-dammarane-6*α*,12*β*,25-triol (**4a**)

White powder (yield 53.1%), ^1^H-NMR (CDCl_3_, 600 MHz) *δ* 5.64 (s, 1H), 4.47–4.44 (m, 1H), 4.08–4.04 (m, 1H), 3.83–3.81 (m, 1H), 3.51–3.47 (m, 1H), 3.16–3.14 (m, 1H), 2.67 (s, 2H), 2.19–2.15 (m, 1H), 2.00–1.97 (m, 2H), 1.90–1.79 (m, 3H), 1.68–1.47 (m, 9H), 1.24 (s, 3×-CH_3_), 1.13 (s, -CH_3_), 1.11–1.09 (m, 1H), 1.07 (s, -CH_3_), 1.03 (s, -CH_3_), 1.02 (s, -CH_3_), 0.97–0.93 (m, 2H), 0.91 (s, 2x-CH_3_). ^13^C-NMR (CDCl_3_, 150 MHz) *δ* 171.99, 86.62, 85.51, 81.92, 70.97, 70.35, 68.26, 61.29, 58.79, 51.99, 49.96, 49.09, 48.01, 47.00, 42.58, 40.98, 39.04, 38.50, 33.81, 32.73, 31.41, 31.31, 30.84, 28.73, 28.06, 27.62, 26.23, 25.11, 18.27, 17.62, 17.06, 16.90, 16.70. HR-MS (ESI, m/z) [M + H]^+^ calcd. for C_33_H_57_NO_6_, 564.4264, found: 564.4269.

(20*S*, 24*R*)-Epoxy-3*β*-*O*-(l-methinyl)-dammarane-6*α*,12*β*,25-triol (**4**b)

Light yellow powder (yield 42.5%), ^1^H-NMR (CDCl_3_, 600 MHz) *δ* 5.62 (s, 1H), 5.39-5.34 (m, 1H), 4.51–4.46 (m, 1H), 3.84–3.81 (m, 1H), 3.62–3.58 (m, 1H), 3.52–3.47 (m, 1H), 2.64–2.58 (m, 2H), 2.08 (s, 2H), 2.07 (s, -SCH_3_), 1.93–1.77 (m, 8H), 1.66–1.50 (m, 10H), 1.33–1.29 (m, 2H), 1.24 (s, 2× -CH_3_), 1.11 (s, -CH_3_), 1.07 (s, -CH_3_), 1.03 (s, -CH_3_), 0.98 (s, -CH_3_), 0.91 (s, -CH_3_), 0.90 (s, -CH_3_). ^13^C-NMR (CDCl_3_, 150 MHz) *δ* 174.73, 86.60, 85.58, 81.24, 72.13, 70.77, 70.28, 58.92, 53.73, 51.96, 50.03, 49.13, 47.95, 47.17, 42.58, 40.95, 39.55, 38.31, 33.78, 33.15, 32.72, 31.32, 30.61, 28.67, 28.02, 27.64, 26.26, 25.08, 23.32, 18.19, 17.47, 17.04, 16.70, 15.47. HR-MS (ESI, m/z) [M + H]^+^ calcd. for C_35_H_61_NO_6_S, 624.4298, found: 624.4293.

(20*S*, 24*R*)-Epoxy-3*β*-*O*-(-l-glycyl)-dammarane-6*α*,12*β*,25-triol (**4c**)

White powder (yield 59.1%), ^1^H-NMR (CDCl_3_, 600 MHz) *δ* 5.62 (s, 1H), 4.52–4.50 (m, 1H), 4.13–4.08 (m, 1H), 3.86–3.83 (m, 1H), 3.52–3.48 (m, 1H), 3.46 (s, 2H), 2.21–2.17 (m, 1H), 2.05–1.96 (m, 3H), 1.93–1.81 (m, 6H), 1.70–1.54 (m, 8H), 1.27 (s, -CH_3_), 1.26 (s, -CH_3_), 1.16 (s, -CH_3_), 1.09 (s, -CH_3_), 1.06 (s, -CH_3_), 1.04 (s, -CH_3_), 0.98 (s, 1H), 0.97 (s, 1H), 0.93 (s, -CH_3_), 0.93 (s, -CH_3_). ^13^C-NMR (CDCl_3_, 150 MHz) *δ* 73.40, 86.48, 85.43, 81.62, 70.80, 70.14, 68.30, 61.29, 51.82, 49.89, 49.00, 47.88, 47.03, 43.73, 40.87, 38.94, 38.38 (×2C), 32.60, 31.25, 31.19, 30.70, 28.58, 27.90, 27.52, 26.13, 24.95, 23.39, 18.12, 17.49, 16.93,16.53. HR-MS (ESI, m/z) [M + H]^+^ calcd. for C_32_H_55_NO_6_, 550.4107, found: 550.4112.

(20*S*, 24*R*)-Epoxy-3*β*-*O*-(l-leucyl)-dammarane-6*α*,12*β*,25-triol (**4d**)

Light yellow powder (yield 34.2%), ^1^H-NMR (CDCl_3_, 600 MHz) *δ* 5.64 (s, 1H), 4.52–4.50 (m, 1H), 4.16–4.11 (m, 1H), 3.88–3.85 (m, 1H), 3.68–3.66 (m, 1H), 3.56–3.51 (m, 1H),2.23–2.19 (m, 1H), 2.06–2.01 (m, 4H), 1.96–1.81 (m, 6H), 1.74–1.52 (m, 10H), 1.29 (s, 2×-CH_3_), 1.27 (s, -CH_3_), 1.19 (s, -CH_3_), 1.11 (s, -CH_3_), 1.09 (s, 2×-CH_3_), 1.01–0.99 (m, 2H), 0.98 (s, -CH_3_), 0.96 (s, -CH_3_), 0.95 (s, -CH_3_). ^13^C-NMR (CDCl_3_, 150 MHz) *δ* 175.85, 86.49, 85.43, 81.32, 70.80, 70.15, 68.37, 61.35, 53.22, 51.83, 49.89, 49.00, 47.89, 47.05, 43.77, 40.89, 38.96, 38.33, 37.11, 32.60, 31.20, 30.77, 29.71, 28.59, 27.90, 27.53, 26.14, 24.96, 24.85, 23.35, 23.00, 21.82, 18.12, 17.44, 16.94, 16.65. HR-MS (ESI, m/z) [M + H]^+^ calcd. for C_36_H_63_NO_6_, 606.4733, found: 606.4738.

(20*S*, 24*R*)-Epoxy-3*β*-*O*-(l-prolyl)-dammarane-6*α*,12*β*,25-triol (**4e**)

White powder (yield 45.8%), ^1^H-NMR (CDCl_3_, 600 MHz) *δ* 5.64 (s, 1H), 5.40–5.35 (m, 1H), 4.57–4.53 (m, 1H), 4.03–4.00 (m, 1H), 3.84–3.81 (m, 1H), 3.51–3.46 (m, 1H), 3.34–3.16 (m, 2H), 2.37–2.24 (m, 3H), 2.16 (s, 1H), 2.04–1.83 (m, 9H), 1.68–1.46 (m, 10H), 1.33–1.30 (m, 2H), 1.24 (s, 3x-CH_3_), 1.09 (s, -CH_3_), 1.07 (s, -CH_3_), 1.00 (s, -CH_3_), 0.97 (s, -CH_3_), 0.90 (s, -CH_3_). ^13^C-NMR (CDCl_3_, 150 MHz) *δ* 171.59, 86.58, 85.52, 82.66, 73.73, 70.72, 70.34, 67.91, 59.74, 51.95, 49.98, 49.09, 47.93, 46.71, 46.37, 42.45, 40.98, 39.59, 38.27, 32.64, 31.33, 30.65, 29.84, 29.54, 28.70, 28.05, 27.59, 26.17, 25.10, 24.98, 24.54, 18.12, 17.47, 16.93, 16.69. HR-MS (ESI, m/z) [M + H]^+^ calcd. for C_35_H_59_NO_6_, 590.4420, found: 590.4423.

(20*S*, 24*R*)-Epoxy-3*β*-*O*-(l-valyl)-dammarane-6*α*,12*β*,25-triol (**4f**)

White powder (yield 41.5%), ^1^H-NMR (CDCl_3_, 600 MHz) *δ* 5.59 (s, 1H), 5.30–5.24 (m, 1H), 4.54–4.52 (m, 1H), 3.87–3.85 (m, 1H), 3.55–3.51 (m, 1H), 3.30 (d, *J =* 4.5 Hz 1H), 2.22–2.18 (m, 1H), 2.12–2.04 (m, 3H), 1.92–1.85 (m, 3H), 1.74–1.55 (m, 8H), 1.51-1.46 (m, 2H), 1.33–1.31 (m, 2H), 1.29 (s, -CH_3_), 1.28 (s, -CH_3_), 1.15-1.13 (m, 1H), 1.11 (s, -CH_3_), 1.09–1.06 (m, 1H), 1.03 (s, -CH_3_), 1.00 (s, -CH_3_), 0.91 (s, -CH_3_), 0.90 (s, -CH_3_), 0.88 (s, -CH_3_), 0.87 (s, -CH_3_). ^13^C-NMR (CDCl_3_, 150 MHz) *δ* 175.20, 86.52, 85.43, 81.36, 70.94, 70.10, 67.03, 60.36, 56.09, 52.02, 50.43, 49.40, 47.97, 39.77, 38.61, 37.92, 37.07, 34.76, 32.62, 31.77, 31.35, 31.21, 28.59, 28.07, 27.91, 27.60, 26.14, 25.00, 23.77, 19.65, 19.21, 16.76, 16.60, 16.35, 15.40. HR-MS (ESI*,* m/z) [M + H]^+^ calcd. for C_35_H_61_NO_6_, 592.4577, found: 592.4572.

### 3.2. Cytotoxic Activity Determination on HepG2 Cells

#### 3.2.1. Cell Culture

Human hepatocellular liver carcinoma cell line (HepG2) were obtained from Institute of Biochemistry and Cell Biology, Academy of Science (Shanghai, China) and were authenticated using the STR profiling method. HepG2 cells were cultured in MDEM (Gibco, Waltham, MA, USA) supplemented with 10% fetal bovine serum (FBS) and 1% penicillin-streptomycin bubbled with 5% CO_2_-enriched air at a temperature of 37 °C (Thermo Scientific, Waltham, MA, USA).

#### 3.2.2. Cell Viability Assay

The cytotoxic effect of compounds on HepG2 cells was assessed by using CCK-8 method [81]. Briefly, HepG2 cells were cultured in 96-well plates at a density of 5 × 10^3^ cells per well for 24 h. Then, the cells were treated with test compounds at different concentrations of 0, 6.25, 12.5, 25, 50, 100, 200 μM which dissolved in Dimethyl sulfoxide (DMSO) and incubated in CO_2_ incubator for 48 h (37 °C, 5% CO_2_). After incubation, 10 μL CCK-8 (5 mg/mL) was added to each well and then incubated for another 4 h, then the OD value of each pore was measured at 450 nm by enzyme labeling instrument. Three replicates were performed. The relative cell viability was expressed as percentage relative to the untreated cells. The inhibition ratio was calculated. The IC_50_ values were calculated using Logistic regression from three independent tests.

### 3.3. Effects of Compound 2e on H22 Tumor-Bearing Mice

#### 3.3.1. Cell Lines and Animals

Hepatocellular carcinoma cell line (H22) was purchased from the Cell Bank of Type Culture Collection Committee of Chinese Academy of Sciences (Shanghai, China), and cultured in Dulbecco’s modified eagle medium supplemented with 10% fetal calf serum and then cultured at 37 °C in 5% CO_2_.

Female and male Kunming mice (SPF grade, weighing 18–22 g) were purchased from Changchun Yisi Experimental Animal Technology Co., Ltd. (Changchun, China). All the mice, given food and water ad libitum, were raised in temperature and humidity controlled (23 ± 2 °C, 50 ± 10% humidity) environment under a 12 h light/dark cycle. The mice were acclimatized for at least 1 week before use. The animal experiments were performed in accordance with the guidelines for the Care and Use of Laboratory Animals and approved by the ethics committee of School of Pharmaceutical Sciences, Jilin University, China.

#### 3.3.2. Materials

Cyclophosphamide (CTX), used as the positive drug, was purchased from Sigma (St. Lousis, MO, USA. Detecting kits for IL-2, TNF-*α*, VEGF, AST, ALT, CRE and BUN were purchased from Nanjing Jiancheng Biotechnology Co. (Nanjing, China).

#### 3.3.3. Tumor-Bearing Mice Model and Drug Administration

H22 cells, resuspended in normal saline with final concentration of 5 × 10^6^ cells/mL, were injected into the abdominal cavity of mice under aseptic conditions. After 1 week, ascites tumor cells were extracted and dispersed with physiological saline to a concentration of 1 × 10^7^ cells/mL.

The diluted ascites tumor cell suspension was then injected (0.2 mL, 2 × 10^6^ cells/mouse) subcutaneously into the right forelimb armpit of mice at day 1 to establish the tumor-bearing mice model. The mice did not injected the tumor cells were used as the normal control.

24 h after transplanting the H22 cells, mice were divided into seven groups (*n* = 10): (i) normal control group (N, physiological saline); (ii) tumor-bearing model group (M, physiological saline); (iii) cyclophosphamide positive drug group (CTX, 20 mg/kg); (iv) low-dose compound **2e** group (L-**2e**, 25 mg/kg; (v) moderate-dose compound **2e** group (M-**2e**, 50 mg/kg); (vi) high dose compound **2e** group (H-**2e**, 100 mg/kg); (vii) compound **2e** (50 mg/kg) + CTX (20 mg/kg) combination group. Mice in each group were intragastrically administered once a day continuously for 14 days. Administration volume were all 20 mL/kg. CTX, L-**2e**, M-**2e**, H-**2e** and **2e** + CTX were respectively dissolved in physiological saline to form the aqueous solution with the corresponding concentrations.

#### 3.3.4. Antitumor Activity Evaluation

Before administration every day, the tumor size and body weight were measured. When the tumor was larger than 20 mm in diameter, animals were euthanized according to the IACUC proposals.

24 h after the last administration, the whole blood was collected from the orbit. Serum was obtained from whole blood by centrifugation (3500 rpm, 15 min, 4 °C) and stored at −20 °C. The levels of IL-2, TNF-*α*, VEGF, AST, ALT, BUN and CRE in serum were determined in accordance with the procedures described in commercial kit instructions.

Mice were then killed by cervical dislocation after blood collected. The tumor, liver and kidney were rapidly separated, weighted and dissected for histopathological examination.

The tumor inhibition rate (TIR) was calculated as follows: TIR (%) = (W_m_ − W_t_)/W_m_ × 100% (W_m_: tumor weight of model group, W_t_: tumor weight of treatment group).

The liver and kidney indexes were calculated as follows: organ index (mg/g) =average weight of organ/average body weight.

The tissue of tumor, liver and kidney from each mouse was quickly fixed in 10% neutral-buffered formalin for histopathology. The H&E staining of H22 solid tumor, liver and kidney tissue were embedded in paraffin and sectioned into sections with a thickness of 5 µm which were conducted following the manufacturer’s instructions in respective kit. Tumor, liver and kidney tissue sections were observed by light microscopy (Leica DM750, Solms, Germany) and recorded by photography (×400).

#### 3.3.5. Statistical Analysis

The data were expressed as mean ± SD. All statistical analyses were performed using the SPSS 16.0 software (SPSS, USA). For statistical comparison of values, a Student’s *t*-test was used, and *p* < 0.05 was considered statistically significant. One-way ANOVA, followed by *t*-test, was used to compare differences among groups.

### 3.4. Metabolomics Study

#### 3.4.1. Materials

From Sigma-Aldrich (St. Louis, MO, USA), formic acid, linoleic acid (purity ≥ 99%) and arachidonic acid (purity ≥ 97%) were purchased. From Fisher Chemical Company (Geel, Belgium), acetonitrile and methanol were obtained. Deionized water was prepared by Millipore water purification system (Millipore, Billerica, MA, USA). Paraxanthine (purity ≥ 7%), Chloral hydrate, acetic acid (36–38%) and l-tryptophan were purchased from Beijing Laiyao Biological Technology Co., Ltd. (Beijing, China), Biosharp Co., Ltd. (Shenyang, China), J&K Technology Co., LTD. (Hong Kong, China) and the National Institutes for Food and Drug Control (Beijing, China), respectively.

#### 3.4.2. UPLC-QTOF-MS Conditions

Waters UPLC system with electrospray ionization (ESI) interface combined with Xevo G2-XS QTOF mass spectrometer was applied for determination and analysis. 

LC conditions were as follows: ACQUITY UPLC BEH C18 (100 × 2.1 mm, 1.7 µm) with column temperature being set as 30 °C; mobile phase consisting of 0.1% formic acid in water (eluent A) and 0.1% formic acid in acetonitrile (eluent B) with the elution conditions (0~min, 10% B; 2–26 min, 10%→90% B; 26–28 min, 90% B; 28–28.1 min, 90%→10% B; 28.1–30 min, 10% B); flow rate at 0.4 mL/min; 10% and 90% acetonitrile aqueous solutions being used as weak and strong wash solvents, respectively; the temperature of sample manager being set at 10 °C.

MS conditions were as follows: desolvation and source temperatures being and 150 °C, respectively; cone and desolvation gas flows being 50 L/h and 800 L/h, respectively; cone voltage setting at 40 V; capillary voltage being at 2.2 kV (ESI-) and 2.6 kV (ESI+); MS^E^ centroid mode with low energy of 6 V and high energy of 20~40 V; sodium formate being used to calibrate the mass spectrometer in the range of 50 to 1200 Da; external reference leucine enkephalin (m/z 556.2771 and 554.2615 in ESI+ and ESI- modes) injected at a flow of 10 µL/min.

QC sample was injected randomly throughout the whole worklist for 5 times. The volume injections of QC and test samples were all 5 µL per run. MassLynx V4.1 workstation (Waters, Manchester, UK) was used to record data.

#### 3.4.3. Metabolomics Study

Firstly, MarkerLynx XS software (Version 4.1, Waters Co., Milford, MA, USA.) was applied to analyze data with the optimized parameters (mass range 50~1200 Da, mass tolerance 0.10 with window 0.10, retention time 2~28 min with window 0.20, noise elimination level 6 and marker intensity threshold 2000 counts). Then the results of exact mass/retention time pairs and the corresponding intensities were shown in Extended Statistics (XS) Viewer. 

Secondly, SIMCA-P software (Version 14.1, Umetric, Umea, Sweden) was used to perform the multivariate analysis including principal component analysis (PCA) and orthogonal projections to latent structures discriminant analysis (OPLS-DA) [34]. S-plots were then created to explore the remarkable potential biomarkers with VIP (variable importance in the projection) value above 1.0 and *p*-value below 0.05. In addition, the permutation test with R^2^/Q^2^ values indicating statistical significance was performed to provide a reference distribution. The predictive ROC (receiver operating characteristic) curves were generated using the metabolites with AUC (area under curve) >0.8 and *p* < 0.01. Afterward, the biochemical databases including HMDB (http://www.hmdb.ca/), METLIN (http://metlin.scripps.edu/), MetaboAnalyst (http://www.metaboanalyst.ca/), and KEGG (http://www.kegg.com/) were applied to further identify the biomarkers by either referring the chemical standards or comparing the MS/MS fragmentation patterns. [M + H]^+^ and [M + Na]^+^ in ESI+, [M − H]^−^ and [M + FA − H]^−^ in ESI-, were used as the adducts with the mass tolerance at 10 ppm.

Finally, MetaboAnalyst 4.0 software was used to filter out the most vital potential metabolic pathways (impact-value threshold above 0.10) by analyzing the confirmed distinct metabolites.

## 4. Conclusions

In this paper, a total of 24 amino acid derivatives, including 20 new along with 4 known compounds of pyxinol and ocotillol were synthesized and evaluated in vitro and in vivo for the anti-hepatocarcinoma effect. Most of the amino acid derivatives showed obvious enhanced activity compared with pyxinol or ocotillol. Compound **2e** displayed excellent activity in HepG2 human cancer cell and in H22 tumor-bearing mice. It was also revealed that compound **2e** combined with cyclophosphamide (CTX) had the best anti-tumor activity and the lowest toxicity in mice. A total of 21 potential metabolites involved in 8 metabolic pathways were identified in anti- hepatocarcinoma effect of compound **2e**. These results suggest that compound **2e** is a promising agent for anti-hepatocarcinoma with low toxicity, and that it also could increase CTX’s efficiency and reduce its toxicity.

## Figures and Tables

**Figure 1 molecules-26-00780-f001:**
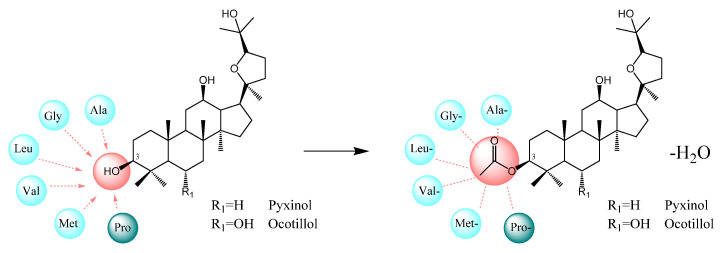
Design of amino acid derivatives of ocotillol-type sapogenins.

**Figure 2 molecules-26-00780-f002:**
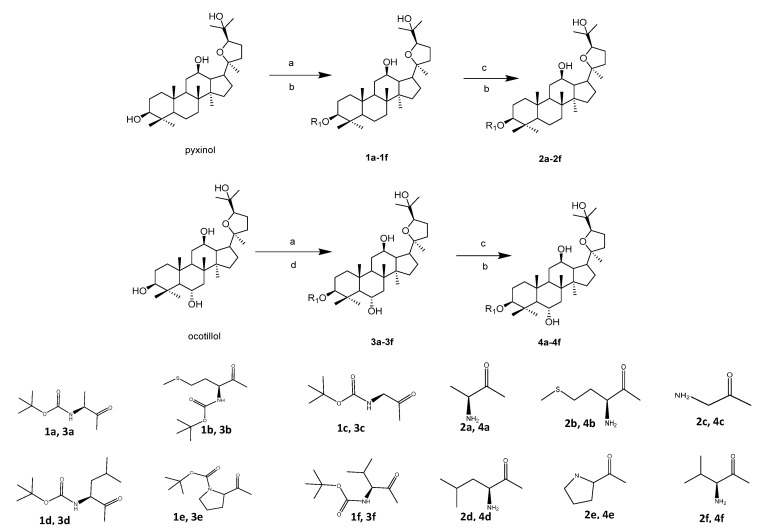
Synthesis of pyxinol and ocotillol derivatives **1a–1f**, **2a–2f**, **3a–3f**, **4a–4f**. Reagents and conditions: (**a**) DMAP, EDC, boc-amino acid, r.t. (**b**) dry CH_2_Cl_2_, r.t. (**c**) TFA, r.t. (**d**)THF, r.t.

**Figure 3 molecules-26-00780-f003:**
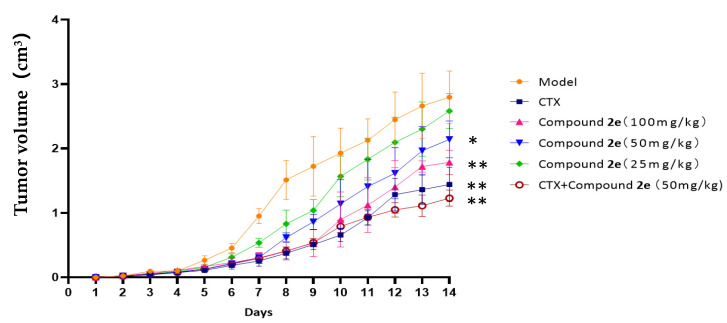
Effect on tumor volume. Compared with the model group, * *p* < 0.05, ** *p* < 0.01.

**Figure 4 molecules-26-00780-f004:**
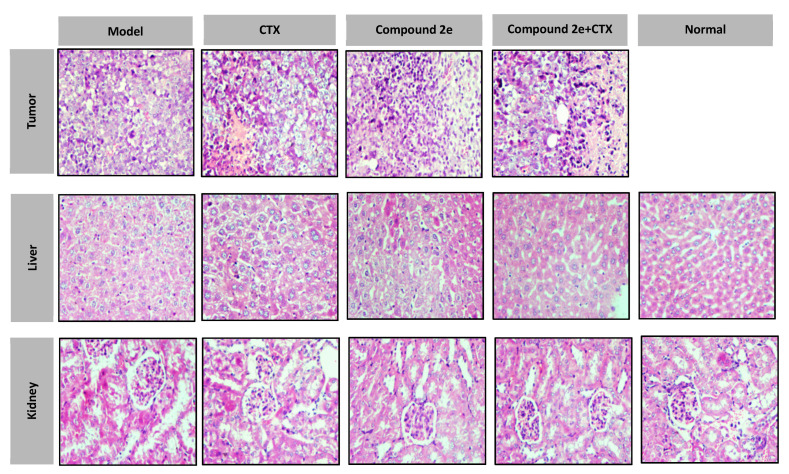
Histological examination of morphological changes in tumor, liver and kidney.

**Figure 5 molecules-26-00780-f005:**
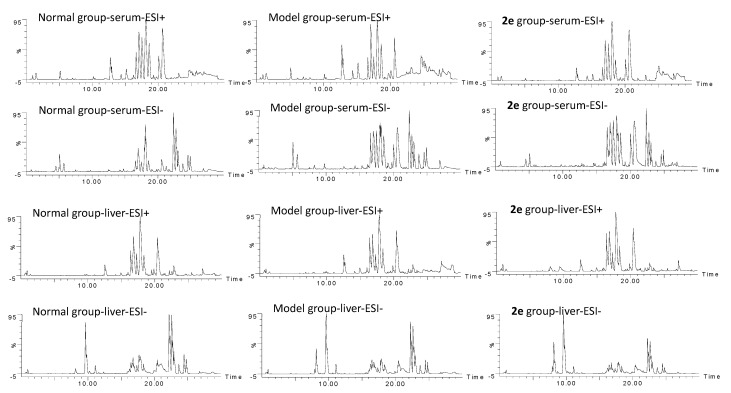
The representative base peak intensity (BPI) chromatograms in ESI+ and ESI– modes.

**Figure 6 molecules-26-00780-f006:**
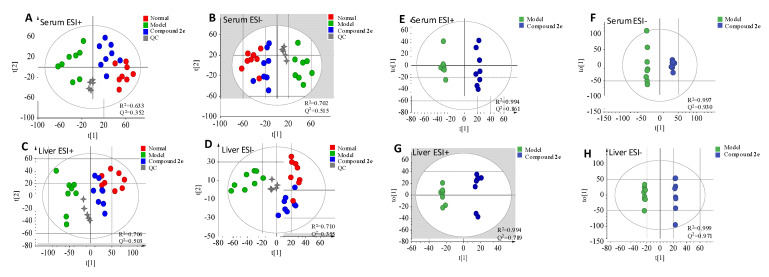
Principal component analysis (PCA) (**A**–**D**) and OPLS-DA (**E**–**H**) score plots of serum and liver metabolic profiling.

**Figure 7 molecules-26-00780-f007:**
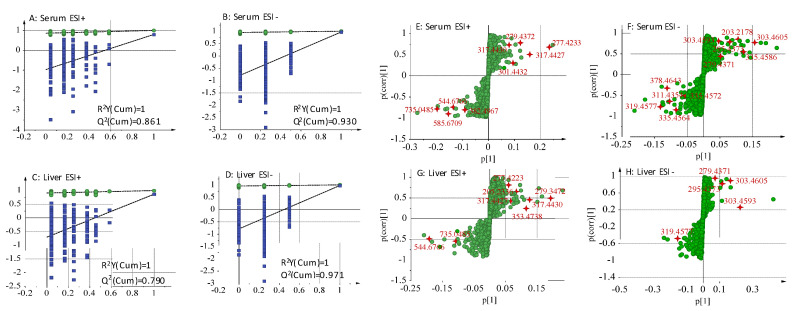
The permutation plots (**A**–**D**) and S-plots (**E**–**H**) of serum and the liver metabolic profiling.

**Figure 8 molecules-26-00780-f008:**
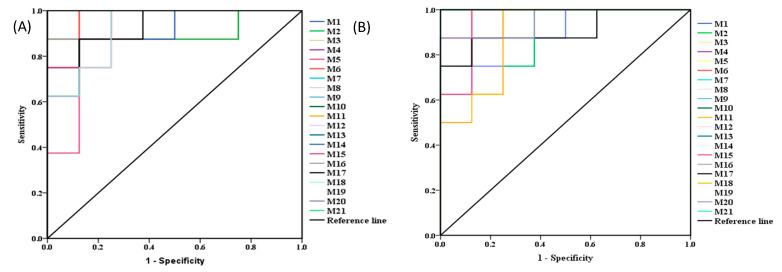
The predictive receiver operating characteristic (ROC) curves generated using 21 biomarkers contributing to (**A**) liver-tumor progress between model group and normal group, (**B**) compound **2e** treatment between model group and compound **2e** group.

**Figure 9 molecules-26-00780-f009:**
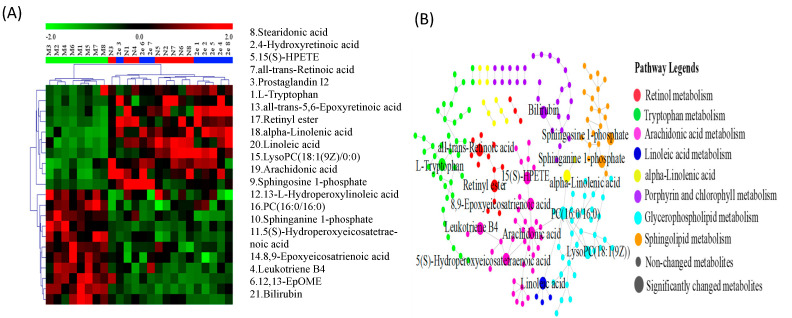
The heatmap (**A**) and the metabolic pathways (**B**) of all potential biomarkers.

**Table 1 molecules-26-00780-t001:** IC_50_ (µM) of all compounds over HepG2 cells.

Compounds	IC_50_ (µM)	Compounds	IC_50_ (µM)
**Pyxinol**	74.49 ± 1.09	**Ocotillol**	92.65 ± 3.07
**1a**	26.33 ± 0.72	**3a**	60.95 ± 1.57
**1b**	73.55 ± 2.60	**3b**	88.60 ± 5.03
**1c**	26.63 ± 1.52	**3c**	32.62 ± 1.55
**1d**	73.14 ± 1.67	**3d**	74.91 ± 2.87
**1e**	30.46 ± 2.98	**3e**	41.75 ± 2.58
**1f**	62.15 ± 3.78	**3f**	80.10 ± 2.03
**2a**	16.46 ± 1.66	**4a**	33.99 ± 0.90
**2b**	20.29 ± 2.04	**4b**	50.65 ± 0.62
**2c**	16.31 ± 0.31	**4c**	20.55 ± 2.47
**2d**	20.13 ± 3.49	**4d**	35.44 ± 1.59
**2e**	11.26 ± 0.43	**4e**	24.62 ± 0.35
**2f**	19.74 ± 1.96	**4f**	44.74 ± 3.07
**Cyclophosphamide**	9.18 ± 0.34	-	-

The values given are mean of three experiments ±SD.

**Table 2 molecules-26-00780-t002:** Effects on body weight, tumor weight and organ indexes.

Groups	Dosages (mg/kg)	BODYWEIGHT (g)		Increase of Body Weight (g)	Tumor Weight (g)	Inhibitory Rate (%)	Liver Index(mg/g)	Kidney Index (mg/g)
		D1	D14					
Normal	-	22.35 ± 1.58	36.59 ± 3.63	14.24 ± 2.61	-	-	37.99 ± 6.03	11.08 ± 1.72
Model	-	21.24 ± 1.66	35.86 ± 3.53	14.62 ± 3.49	1.61 ± 0.35	-	45.69 ± 6.19	13.89 ± 2.24
CTX	20	21.20 ± 1.71	32.76 ± 3.14	11.56 ± 3.23	0.81 ± 0.18 **	49.69%	51.54 ± 7.69	13.27 ± 1.96
Compound **2e**	25	23.12 ± 2.18	35.27 ± 3.18	12.15 ± 3.71	1.35 ± 0.26	16.20%	42.89 ± 6.04	12.24 ± 1.92
	50	22.45 ± 1.89	34.99 ± 3.22	12.54 ± 2.72	1.21 ± 0.34 *	24.76%	42.77 ± 4.55	12.61 ± 1.42
	100	22.51 ± 1.70	35.64 ± 2.89	13.13 ± 3.36	1.04 ± 0.23 **	35.32%	41.83 ± 4.31	12.42 ± 1.21
**2e** + CTX	50 + 20	23.07 ± 2.29	37.65 ± 3.70	14.58 ± 2.68	0.72 ± 0.20 **	55.30%	37.01 ± 4.47 **	11.72 ± 2.24 *

Compared with the model group, * *p* < 0.05, ** *p* < 0.01. Mice were intragastrically administered once a day continuously for 14 days.

**Table 3 molecules-26-00780-t003:** Effects of compound **2e** on cytokine levels (mean ± SD, *n* = 6) in H22 tumor-bearing mice.

Groups	Dosages (mg/kg)	TNF-*α* (pg/mL)	IL-2 (pg/mL)	VEGF (pg/mL)
Normal	-	306.77 ± 17.47	150.93 ± 6.01	97.76 ± 13.75
Model	-	388.35 ± 28.22 ^##^	95.81 ± 16.50 ^##^	188.94 ± 22.43 ^##^
CTX	20	207.47 ± 23.37 ^##,^ **	110.52 ± 17.54 ^##^	103.43 ± 14.00 **
Compound **2e**	25	287.01 ± 39.10 **	111.13 ± 13.28 ^##^	158.32 ± 15.97 ^##,^ *
	50	321.97 ± 55.57 *	136.86 ± 22.66 **	127.62 ± 17.46 ^##,^ **
	100	358.35 ± 45.19 ^#^	174.11 ± 28.90 **	114.67 ± 15.42 **
**2e** + CTX	50 + 20	276.48 ± 26.89 ^#,^ **	214.81 ± 26.20 ^##,^ **	109.86 ± 22.90 **

Compared with the normal control group, ^#^
*p* < 0.05, ^##^
*p* < 0.01; Compared with the model group, * *p* < 0.05, ** *p* < 0.01.

**Table 4 molecules-26-00780-t004:** Effects of compound **2e** treatment on hepatic and renal function.

Groups	Dosages (mg/kg)	ALT (IU/L)	AST (IU/L)	CRE (μmol/L)	BUN (mmol/L)
Normal	-	19.48 ± 2.36	24.78 ± 3.38 **	20.71 ± 7.77	9.09 ± 1.81
Model	-	73.60 ± 2.76 ^##,^ **	91.8 ± 6.62 ^##,^ **	41.89 ± 10.27 ^##^	20.75 ± 4.96 ^##^
CTX	20	50.61 ± 3.22 ^##,^ **	53.47 ± 5.04 ^##,^ **	32.86 ± 7.74 ^#^	14.23 ± 2.38 ^##,^ *
Compound **2e**	25	43.62 ± 5.15 ^##,^ **	42.51 ± 5.22 ^##,^ **	31.17 ± 9.03	12.70 ± 2.24 ^#,^ **
	50	42.31 ± 5.00 ^##,^ **	38.69 ± 6.43 ^##,^ **	28.95 ± 9.16 *	11.06 ± 2.16 **
	100	35.37 ± 3.32 ^##,^ **	31.47 ± 4.48 ^#,^ **	26.93 ± 10.60 *	10.32 ± 2.77 **
**2e** + CTX	50+20	39.95 ± 7.21 ^##,^ **	40.17 ± 6.27 ^##,^ **	30.45 ± 8.62	12.66 ± 3.06 ^#,^ **

Compared with the normal control group, ^#^
*p* < 0.05, ^##^
*p* < 0.01; Compared with the model group, * *p* < 0.05, ** *p* < 0.01.

**Table 5 molecules-26-00780-t005:** The RSDs (%) of peak intensity and retention time in validation test.

	Serum				Liver			
	Peak Intensity		Retention Time		Peak Intensity		Retention Time	
	ESI+	ESI-	ESI+	ESI-	ESI+	ESI-	ESI+	ESI-
Stability of LC-MS system	1.56–5.57	1.02–4.98	0.23–2.35	0.64–2.74	1.97–7.14	1.34–6.61	0.63–3.22	0.89–2.71
Injection Precision	0.92–3.09	1.06–5.14	0.66–2.48	0.97–3.05	1.36–6.47	2.14–6.83	0.74–3.53	0.98–3.33
Reproducibility of sample preparation	2.12–6.69	1.68–6.57	1.02–4.39	1.25–3.35	2.77–5.61	2.19–5.88	1.30–2.94	1.28–4.38
Post-preparation stability of sample	2.39–6.41	2.01–5.27	1.23–4.18	1.36–3.99	1.44–4.34	1.67–5.25	1.33–3.36	1.41–3.79

**Table 6 molecules-26-00780-t006:** Distinct metabolites identified in serum and liver samples.

No.	tR/min	Measured Mass (Da)	VIPvalue	Formula	Mass Error (ppm)	Adducts	Biomarkers	HMDB ID	Pathway	Content Level	Source
1 *	1.51	203.2178	5.25	C_11_H_12_N_2_O_2_	3.13	M − H	l-Tryptophan	0000929	TrpM	C_M_ < C_2e_ < C_N_	Serum
	1.49	205.2336	2.84		2.21	M + H					Liver
2 ^a^	10.1	317.4430	4.34	C_20_H_28_O_3_	1.45	M + H	4-Hydroxyretinoic acid	0006254	RM	C_M_ < C_2e_ ≈ C_N_	Serum
			4.66			M + H					Liver
3 ^a^	10.19	353.4738	7.20	C_20_H_32_O_5_	2.13	M + H	Prostaglandin I2	0001335	AM	C_M_ < C_N_ < C_2e_	Liver
4 ^a^	14.06	335.4564	4.17	C_20_H_32_O_4_	–3.87	M − H	Leukotriene B4	0001085	AM	C_M_ > C_2e_ ≈C_N_	Serum
5 ^a^	14.58	335.4586	7.36	C_20_H_32_O_4_	2.62	M − H	15(S)-HPETE	0004244	AM	C_M_ < C_N_<C_2e_	Serum
6 ^a^	14.59	295.4373	4.04	C_18_H_32_O_3_	1.19	M − H	12,13-EpOME	0004702	LM	C_M_ > C_2e_ ≈ C_N_	Serum
			3.57			M − H					Liver
7 ^a^	14.61	301.4432	6.29	C_20_H_28_O_2_	0.37	M + H	all-trans-Retinoic acid	0001852	RM	C_M_ < C_N_ < C_2e_	Serum
8 ^a^	14.64	277.4223	12.73	C_18_H_28_O_2_	2.11	M + H	Stearidonic acid	0006547	ALAM	C_M_ < C_2e_ ≈ C_N_	Serum
			2.44			M + H					Liver
9 ^a^	15.38	378.4643	8.27	C_18_H_38_NO_5_P	1.21	M − H	Sphingosine 1-phosphate	0000277	SM	C_M_ > C_N_ > C_2e_	Serum
10 ^a^	16.02	382.4967	5.18	C_18_H_40_NO_5_P	2.87	M + H	Sphinganine 1-phosphate	0001383	SM	C_M_ > C_2e_ ≈ C_N_	Serum
11 ^a^	16.28	335.4572	3.18	C_20_H_32_O_4_	−1.4	M − H	5(*S*)-Hydroperoxyeicosatetraenoic acid	0001193	AM	C_M_ >C_2e_ ≈ C_N_	Serum
12 ^a^	16.93	311.4354	5.29	C_18_H_32_O_4_	−2.97	M − H	13-l-Hydroperoxylinoleic acid	0003871	LM	C_M_ > C_2e_ ≈ C_N_	Serum
13 ^a^	17.74	317.4427	11.07	C_20_H_28_O_3_	0.53	M + H	all-trans-5,6-Epoxyretinoic acid	0012451	RM	C_M_ < C_2e_ ≈ C_N_	Serum
	17.49		2.14			M + H					Liver
14 ^a^	18.13	319.4577	7.58	C_20_H_32_O_3_	−1.78	M − H	8,9-Epoxyeicosatrienoic acid	0002232	AM	C_M_ > C_2e_ > C_N_	Serum
			2.19			M − H					Liver
15 ^a^	18.24	544.6766	8.01	C_26_H_52_NO_7_P	2.43	M + Na	LysoPC(18:1(9Z)/0:0)	0002815	GlyM	C_M_ > C_N_ > C_2e_	Serum
			10.90		−1.32	M + H					Liver
16 ^a^	20.35	735.0485	10.77	C_40_H_80_NO_8_P	2.3	M + H	PC(16:0/16:0)	0000564	LM,AM,GlyM,ALAM	C_M_ > C_2e_ ≈ C_N_	Serum
			4.89			M + Na					Liver
17 ^a^	21.3	303.4593	2.20	C_20_H_30_O_2_	1.32	M − H	Retinyl ester	0003598	RM	C_M_ < C_2e_ < C_N_	Serum
			12.45			M − H					Liver
18 ^a^	21.27	279.4372	6.65	C_18_H_30_O_2_	−1.23	M + H	alpha-Linolenic acid	0001388	ALAM	C_M_ < C_2e_ < C_N_	Serum
			6.64			M + H					Liver
19 *	22.8	303.4605	8.16	C_20_H_32_O_2_	5.21	M − H	Arachidonic acid	0001043	AM	C_M_ > C_2e_ > C_N_	Serum
			4.95			M − H					Liver
20 *	23.08	279.4371	2.78	C_18_H_32_O_2_	−1.35	M − H	Linoleic acid	0000673	LM	C_M_ < C_N_ < C_2e_	Serum
			2.15			M − H					Liver
21 ^a^	26.03	585.6709	8.30	C_33_H_36_N_4_O6	1.32	M + H	Bilirubin	0000054	PCM	C_M_>C_N_>C_2e_	Serum

* Metabolites validated with standards. ^a^ Metabolites confirmed by MS/MS fragments. C_M_: content in the model group; C_N_: content in normal group; C**_2e_:** content in **2e** group.

**Table 7 molecules-26-00780-t007:** The results from metabolic pathways of differential metabolites.

Pathway Name	Match Status	*p*-Value	-log (*p*)	Holm *p*	FDR	Impact
Arachidonic acid metabolism (AM)	24/36	<0.0001	9.5736	<0.0001	<0.0001	0.7712
Linoleic acid metabolism (LM)	5/5	<0.0001	3.6273	0.0191	0.0066	1.0000
Retinol metabolism (RM)	7/16	0.0192	1.7123	1.0000	0.4073	0.6347
Sphingolipid metabolism (SM)	6/21	0.1913	0.7183	1.0000	1.0000	0.3185
Alpha-linolenic acid metabolism (ALAM)	3/13	0.4586	0.3386	1.0000	1.0000	0.3333
Glycerophospholipid metabolism (GlyM)	5/36	0.8408	0.075	1.0000	1.0000	0.4032
Porphyrin and chlorophyll metabolis (PCM)	4/30	0.8486	0.071	1.0000	1.0000	0.2986
Tryptophan metabolism (TryM)	1/41	0.9998	<0.0001	1.0000	1.0000	0.1430

**Table 8 molecules-26-00780-t008:** The area under curve (AUC) values and *p*-values of the biomarkers in two predictive ROC curves.

No.	M and N		M and Compound 2e		No.	M and N		M and Compound 2e	
	AUC	*p*-Value	AUC	*p*-Value		AUC	*p*-Value	AUC	*p*-Value
**1**	1	0.001	0.953	0.001	**12**	0.984	0.002	1.000	0.002
**2**	0.906	0.020	0.875	0.015	**13**	1.000	0.001	1.000	0.001
**3**	0.953	0.001	0.984	0.001	**14**	0.922	0.001	0.891	0.004
**4**	0.938	0.001	0.969	0.001	**15**	0.938	0.003	0.953	0.001
**5**	0.891	0.010	0.938	0.001	**16**	0.953	0.003	0.969	0.001
**6**	1.000	0.001	1.000	0.001	**17**	0.922	0.002	0.906	0.005
**7**	1.000	0.001	1.000	0.001	**18**	0.922	0.001	0.891	0.006
**8**	0.969	0.001	0.938	0.002	**19**	0.953	0.001	0.891	0.019
**9**	0.938	0.002	1.000	0.001	**20**	0.906	0.014	0.953	0.005
**10**	0.953	0.003	1.000	0.002	**21**	0.891	0.008	1.000	0.021
**11**	0.953	0.002	0.969	0.001	--	-	-	-	-

M: model group; N: normal group; **2e**: compound **2e** group.

## Data Availability

The data presented in this study are available on request from the corresponding author.

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
