# Peer review of "Synthesis and Anti-Hepatocarcinoma Effect of Amino Acid Derivatives of Pyxinol and Ocotillol"

_molecules, 2021, doi:10.3390/molecules26040780_

Round 1

Reviewer 1 Report

Comments on Manuscript ID “Synthesis and Anti-hepatocarcinoma Effect of Amino Acid Derivatives of Pyxinol and Ocotillol” by Zhang et al.

The authors studied the amino acid derivitization of Pyxinol and Ocotillol for anti-cancer activity with metabolite biomarker validation.

I have following recommendations before the publication of this MS, which need to be addressed in MS:

1.      Line 54: Author wrote-because of poor water solubility, low oral bioavailability and affected absorption in organisms-these sapogenins need to be modified into derivatives with better pharmacological activity- These claims are not supported by any data shown in the MS-The PK/PD mice study would be nice to show and validate the hypothesis of the MS.

2.       Line 86-89, author wrote various advantages of aa (amino acid) derivitization-should be written in introduction after the purpose of aa modification-after line 54. The reason of chosing certain aa should be explained in order to know the concept or advantage of this modification? As aa mostly L-enantiomers are more prone to proteolysis-that is why PK/PD study will give idea about their stability in the animal study.

3.      Table 2: The tumor size with the treatment throughout the experiment would be nice to show as a figure in order to compare the efficacy of the compound 2e or combination with CTX in terms of anti-cancer activity rather than showing the end point tumor weight. Also, highlighting the dosage-frequency and route of administration in the table footnote would be nice to understand the experiment.

4.      Line 181-195-metabolomics study including Fig.4- it would be good if author re-phrase this section regarding what is the purpose of this experiment and what are we comparing here-should be given as one above the other so that comparison of plus and minus chromatograms will be easy to compare. what is meant by ESI+ and ESI- models, need to be explained in the MS?

5.      Similarily, Figure 5 and 6 are not well explained what are they showing and how one can understand their relevance with the study?

6.      All the abbreviations should be defined at their first place in the text-e.g. line 60-define PPD; line 171:AST and ALD?

The paper is technically sound and well written. comments above- will make the MS better.

Reviewer 2 Report

Here are some comments to the text:

-Check the entire document for text/editing errors, for example, the use of italics in scientific names (Salvia barrelieri, Betula humilis, etc), and "in vitro" and "in vivo". -Table 6, check figure foot, "etabolites" in line 285.

-The first part of results can be improved. Particularly, chemistry section, in the first paragraph where synthesis is described, lines 95-106. The characterization and signals shown in the spectra of one of the new derivatives could be discussed. -The figure 2, is missing the -NH2 group in the structure of 2a and 4a.

Reviewer 3 Report

The manuscript describes synthesis of amino acid derivatives of pyxinol and ocotillol and their anti-hepatocarcinoma effect. Both in vitro and in vivo results are interesting. The cytotoxicity potency of most of the amino acid derivatives were stronger than parent compound i.e., pyxinol and ocotillol. In vivo efficacy of 2e significantly inhibited the growth of H22 transplanted tumor. I would like to recommend the manuscript for publication in molecules after minor revision.

Comments –

  1. Among the 24 amino acid derivatives described on the manuscript four are known compounds (2a, 2c, 2d and 2f). Since the amino acid derivatives are already reported previously it is better to use word new rather novel in abstract (page 1, line 10) and throughout the manuscript.
  2. Figure 1 is confusing, R1 group need to be more clearer (position of ester bond), NH2 group is missing in 2a and 2b.
  3. Page 6, lines 186-195, move the description on experimental section.
  4. Figure 4 – hard to compare the chromatogram between Normal, Model and 2e group (either positive or negative ionization mode) because of their position. It will be easier to compare if three sets of data (serum and liver) presented in one column i.e., Normal group-serum-ESI+, Model group-serum-ESI+ and 2e group group-serum-ESI+.
  5. Numbering is double in Reference 31, 40-42, 72.

Round 2

Reviewer 1 Report

The author revised the MS as per all the reviewer's suggestions. Please put the error bars in figure 3 (tumor volume fig.).